# Carbon Flux Explorer Optical Assessment of C, N and P Fluxes

Hannah L. Bourne[1], James K.B. Bishop[1,2], Todd J. Wood[2], Timothy J. Loew[2] and Yizhuang Liu[1]

[1]Department of Earth and Planetary Sciences, University of California, Berkeley, Berkeley, 94720, USA
[2]Lawrence Berkeley National Laboratory, Berkeley, 94720, USA

*Correspondence to*: H. L. Bourne (hbourne@berkeley.edu)

**Abstract.** The magnitude and controls of particulate carbon exported from surface waters and its remineralization at depth are poorly constrained. The Carbon Flux Explorer (CFE), a Lagrangian float-deployed imaging sediment trap, has been designed to optically measure the hourly variations of particle flux to kilometer depths for months to seasons while relaying data in
near-real time to shore via satellite without attending ships. The main optical proxy of particle load recorded by the CFE, volume-attenuance (VA; units of mATN-cm$^2$), while rigorously defined and highly precise, has not been robustly calibrated in terms of particulate organic carbon (POC), nitrogen (PN), and phosphorus (PP). In this study, a novel 3D printed particle sampler using cutting edge additive manufacturing was developed and integrated with the CFE. Two such modified floats (CFE-Cals) were deployed a total of 15 times for 18-24-hour periods to gain calibration imagery and samples at depths near
150 meters in four contrasting productivity environments during the June 2017 California Current Ecosystem – Long Term Ecological Research (LTER) process study. Regression slopes for VA: POC and VA:PN (units mATN-cm$^2$: mmol; $R^2$, n, p-value in parentheses) were $1.01 \times 10^4$ (0.86, 12, <0.001), $1.01 \times 10^5$ (0.86, 15, <0.001) respectively and was not sensitive to particle size classes or the contrasting environments encountered. PP was not well correlated with VA, reflecting the high lability of P relative to C and N. The Volume Attenuance Flux (VAF) to POC flux calibration is compared to previous
estimates.

## 1 Introduction

Marine phytoplankton account for about half (or 50 Pg C y$^{-1}$) of global primary productivity and live for one week on average before being consumed by zooplankton (Falkowski et al., 1998). Approximately 10 Pg C y$^{-1}$ is exported from the surface layer as sinking aggregates containing both particulate organic and inorganic carbon (POC and PIC). The carbon that reaches the
deep ocean remains isolated from the atmosphere for centuries. This process, the "biological carbon pump" (BCP), is a fundamental player in the global carbon cycle, yet the stability of the BCP and its future in the face of climate forced circulation changes and ocean acidification are currently unknown. Recent studies have noted discrepancies in reconciling meso- and bathypelagic activity with current euphotic zone flux estimates (Banse, 2013; Burd et al., 2010; Ebersbach et al., 2011; Passow, 2012; Stanley et al., 2012). Furthermore, estimates of carbon flux out of the euphotic zone range widely from 6 to 12 Pg C y$^{-1}$
(Dunne et al., 2005; Siegel et al., 2014; Yao and Schlitzer, 2013).  More traditional methods of measuring particle flux in the ocean rely on sediment traps or geochemical sampling that require ship time (Buesseler et al., 2007).  As ship time is expensive

both in terms of funding and labor, flux measurements conducted this way are temporally and spatially limited. In recent years, there have been a number of developments towards autonomous instruments capable of measuring particle flux using optical methods (Bishop et al., 2004, 2016; Briggs et al., 2011; Estapa et al., 2013, 2017).

The attenuation of light by particles has long been used by oceanographers as a measurement of particle concentration in the
ocean water column, beginning with development of underwater transmissometers in the early 1970s (Zaneveld, 1973). Transmissometer beam attenuation coefficient (at 660 or 650 nm) has been shown to strongly correlate with measurements of particulate organic carbon (POC) concentration in the water column (Bishop et al., 1999; Bishop and Wood, 2008; Boss et al., 2015; Gardner et al., 2000). Transmissometers were first deployed vertically mounted on Lagrangian profiling floats (called the Carbon Explorers, CEs) in 2001 in the North Pacific (Bishop et al., 2002). These deployments revealed a systematic loss
of transmission as the CEs drifted at depth between profiles. A trend of increasing transmission was seen in the deepest 200-300 m as the CEs rose from 1000 m to the surface, implying that particles had accumulated on the upward looking transmissometer window during drift and were being washed off during initial stages of profiling. CE's deployed in the Southern Ocean in 2002 were modified to measure transmittance before and after exhaust flow from the float's CTD pump was used to clean particles off the transmissometer window during drift and thus a Carbon Flux Index (CFI) was derived as a
systematic measure of particle flux over time (Bishop et al., 2004, Bishop and Wood, 2009). Estapa et al. (2017) advanced the quantitative use of float-deployed transmissometers to estimate particulate carbon flux and more properly derived a flux proxy based on beam attenuance change over the 1-2 days that their neutrally buoyant traps drifted at depth. The Estapa et al. (2017) method does not involve optics flushing.

The Carbon Flux Explorer (CFE), which combines an imaging Optical Sedimentation Recorder (OSR) and profiling Sounding
Ocean Lagrangian Observer (SOLO) float, periodically images particles as they accumulate on a glass sample stage. It thus builds upon the concept of optically measuring particle flux by quantifying particle attenuance at each pixel (Bishop et al., 2016). The imaging instrument also fully resolves particle classes from 20 μm to cm scale. As transmissometer beam attenuation coefficient was found to be highly correlated to POC concentration within the upper 1000 m of the ocean, a reasonable assertion would be that light attenuance of particles integrated across an image (volume attenuance) would also be
highly correlated to POC load. Image attenuance (ATN) is the combined effect of both light scattering loss and light absorption by particles, it is calculated by taking the $-\log_{10}$ of a transmitted light image normalized by an *in-situ* blank composite image of the particle free sample stage (Bishop et al., 2016). Integration of ATN across the sample stage area yields Volume Attenuance (VA, units: mATN-cm$^2$), a measure of particle load. Normalizing by trap opening and time deployed yields Volume Attenuance Flux (VAF, units: mATN-cm$^2$ cm$^{-2}$ d$^{-1}$).

Successful calibration of VAF in carbon units would allow for far greater temporal, and spatial resolution of carbon export than possible with ships and thus inform current models of biological carbon flux as CFEs have the capability of observing hourly variation of particle flux at depth for months to seasons (Bishop et al., 2016). An earlier attempt to calibrate the CFE in

2013 used a surface tethered OSR and sampler (shown in Mcdonnell et al., 2015 Fig. 3F). This method failed as it was discovered that simultaneously deployed surface-tethered OSRs and Lagrangian CFEs collected far different particle types, size distributions, and quantities of material (Bishop et al., 2016). The surface-tethered OSR was biased low due to "baffle bounce" as much as a factor of 20 and collected almost no material larger than 1.5 mm; in other words, the larger aggregates

encounter the cm sized openings of the tethered trap in a near horizontal trajectory and thus bounce back into the flow rather than accumulating in the trap. The samples from the surface tethered OSR were analyzed for phosphorus as a POC proxy, but this approach was not productive as will be shown below. Lacking appropriate calibration samples, Bishop et al. (2016) utilized aggregate size – POC weight estimates from Bishop et al. (1978) to derive a factor of 2.8 for scaling VAF (mATN-cm$^2$ cm$^{-2}$ d$^{-1}$) to POC flux (mmol C m$^{-2}$ d$^{-1}$); they note that applying the Alldredge (1998) volume-POC formula for marine snow particles

collected by scuba in shallow waters yielded a conversion factor of 0.16, about 17 times smaller than the estimate based on Bishop et al. (1978).

Estapa et al. (2017) working in oligotrophic waters near Bermuda compared sediment trap POC flux with transmissometer (650 nm) attenuance drift; conversion of their results (Fig. 7 in Estapa et al., 2017) in our units of VAF: POC yielded factors ranging from 0.46 to 0.74, four to six times lower than the 2.8 conversion factor. We note that a $\log_{10}$ to ln conversion error in

Estapa et al., 2017 implied a greater difference. Multiple optical reasons for differences may include: (1) beam collimation (CFE uses a diffuse LED light source and camera (Bishop et al. 2016) whereas transmissometers are highly collimated but can vary a factor of two in sensitivity based on differences of beam geometry and receiver acceptance angle (Bishop and Wood, 2008; Estapa et al., 2017), (2) effects of particle size distribution on attenuance, (3) wavelength dependence of attenuance (CFE uses the green image plane (~550 nm) vs. (650 nm) red transmissometer), and (4) stray light. Estapa assumed 100%

collection efficiency of particles on the vertically facing transmissometer window and zero contribution of optics biofouling to her measurements. The difference in slopes may be also method dependent as Estapa et al. (2017) analyzed only the particulate carbon in 350 μm screened material from the sediment traps whereas the Bishop et al. (2016) factor includes larger aggregates up to cm size.

Given the finding of a factor of 20 under collection of samples by the surface-tethered OSR, the great uncertainty of literature-

based calibration factors, the few environments sampled, and the multitude of lighting and methodological factors affecting the relationship of attenuance and carbon, we needed to develop a particle sampling device which could operate on the CFE. The new integrated system is referred to as "CFE-Cal" (Fig. 1a, Fig. A4).

Below we describe important design advances that led to the CFE-Cal and report first results from 2 CFE-Cals that were deployed and recovered 15 times at four locations during the June 2 to July 1 2017 California Current Ecosystem Long Term

Ecological Research (CCE-LTER) process study cruise aboard R/V *Revelle*. The aim of the CCE-LTER expedition was to characterize food web processes and particle export at different places within and outside of an offshore-flowing

phytoplankton-rich filament of upwelled water near Point Conception, CA (Fig. 1b). The diverse environments sampled provided an excellent opportunity to collect a calibration sample dataset under high to low particle flux conditions.

## 2 Materials and Methods

### 2.1 CFE, CFE-Cal and Optical Attenuance

Bishop et al. (2016) describe in detail the CFE and the operation of its particle flux sensing OSR. These core elements are identical to those of the CFE-Cal. Briefly, once released from the ship the CFE dives repeatedly below the surface to obtain OSR observations at up to three target depths as it drifts with currents. The CFE's OSR awakes when the target depth is reached. Particles settle through a hexagonal celled baffle (1 cm opening) into a high-aspect ratio funnel (15.4 cm diameter opening) assembly before depositing on a 2.54 cm diameter glass sample stage. Particles are imaged at 13 μm pixel resolution

in three lighting modes: transmitted, transmitted–cross polarized, and dark field. Here we focus on CFE-Cal results and on the calibration of volume attenuance (VA) determined from transmitted light imagery in terms of POC, PN, and PP sample loading. This study focuses on the calibration of VA: POC and PN. In future studies, cross polarized photon yield measured by the CFE, as discussed in Bishop et al. (2016) will be calibrated in terms of PIC units also using the sampler. As the samples collected on filters had large amounts of residual sea salt, the separation of the non-salt Ca requires very high accuracy and a

separate protocol.

On first wake-up of a given CFE dive, the sample stage is flushed with water and images of the particle-free stage are obtained. At timed intervals (~25 min in data described here) the OSR repeats image sets, which register the sequential buildup of particles. The 25-minute interval was determined to be consistent with previous CFE studies (Bishop et al., 2016). After a

predetermined number of image sets over ~1.8 h, stage cleaning occurs and a new reference image set is obtained. After ~5-6 h at a target depth, the OSR performs a final image set, cleaning cycle and reference image set, and the CFE surfaces to report GPS position, CTD profile data and OSR engineering data, and dives to its next target depth. All target depths in this study were chosen to be at 150m. We describe in detail below the particle sampler and its integration with the CFE to form the CFE-Cal. In the case of the CFE-Cal, stage cleaning operations direct particles from each dive to a unique sample bottle.

Image attenuance was calculated following Bishop et al. (2016). Briefly, transmitted light images were normalized by a composite in-situ image of the particle free sample stage. The $-\log_{10}$ of the normalized image was taken to yield attenuance (ATN) values. Pixels with an ATN value less than 0.02 were defined to be background. Pixels with attenuance values above 0.02, determined to be particles, were integrated across the sample stage then divided by total number of pixels in the sample stage area yield average attenuance. This is multiplied by 1000 to yield mATN and then by the sample stage area to give

sample Volume Attenuance (VA, units: mATN-cm$^2$). As light is reduced exponentially as it passes through particles, as long as the overlapping particles do not 100% obscure the transmitted light, attenuance affects are additive. In our analysis, the

transmitted light even in the presence of multiple overlaid large aggregates, never went to zero (in other words, attenuance was never saturating). Therefore, overlapping particles are not an issue in this study.

Depth seeking performance of the CFE-Cal, imaging and sampling times and derived VA time series are illustrated in Figure A1. In order to compare VA to filter loads of POC, PN and PP, the cumulative VA over the course of a dive had to be calculated. The tilt of CFE-Cals during drift is about 3°; the minimization of tilt is required to ensure optimal distribution of particles across the sample stage (Figure A2). During a dive, particles are transferred from image stage to a specific sample bottle between 2 to 6 times. For each cleaning cycle, the VA of a clean image was subtracted from the image with particles prior to transfer to a bottle. This then represented the amount of material directed into the sample bottle after cleaning. VA from each cleaning step was then summed to yield a cumulative VA which should correspond exactly to the particles directed into the sample bottle (Table S1).

## 2.2 Sampler

Most key components of the sampler for the CFE-Cal were fabricated in the Advanced Prototyping Lab at the Jacobs Institute for Design Innovation at UC Berkeley using a Multi-Material Color Objet260 Connex3 (Stratasys, Israel); some parts were also fabricated using the Carbon model M1 3D printer (Redwood City, CA). We chose these particular additive manufacturing processes because they were fast, low-cost, and enabled improved functional designs that were impossible to machine.

The new sampler incorporates the operation and water flow logic of a sampler built in 2004 for our surface buoy-tethered OSR but improves on it considerably (Fig. 7F in McDonnell et al. 2015; Bishop et al., 2016). The physical layout of the sampler is entirely new as the CFE-Cal had to meet stringent dimensional, buoyancy, compressibility, drag performance, and tilt criteria. Furthermore, as the sampler is intended to collect samples for particulate carbon, nitrogen, phosphorous, calcium carbonate, silica, and trace metals, it needed to be non-contaminating.

Figure 2a shows detail of the integration of the sampler with 4 mounted sample collection bottles on the CFE-Cal; Figs. 2b and 2c detail the particle isolation system within each sample bottle. A planetary gear motor (2842S024C; Faulhaber Group, Micromo, Florida) and related custom electronics which actuate the sampler are housed in a pressure compensated acrylic tube filled with Fluorinert (3M) fluid and is mated coaxially with the rotor (Fig. 2d). Fluorinert was selected as it is clear (necessary as there was an optical encoder in the pressure compensated chamber), low viscosity (for motor immersion) and inert (necessary as there were electronics in the chamber). The optical encoder provides feedback as to the proper location for the desired sampling bottle. Figure A3 shows details of the design of key 3D printed elements of the sampler. The sampler inlet is connected to the particle settling stage by a 40 cm long 9.5 mm inner diameter (ID) polyethylene tube (seen in Fig. 2a) and its

outlet is connected by a second 20 cm polyethylene tube to a SBE Model 5T (2000 RPM) pump (Sea Bird Electronics, WA). Flow rate during cleaning was ~ 20 mL s$^{-1}$. When the CFE reaches depth on a new dive, the rotor is moved to select a water path that bypasses the sample bottles (Fig. A2, port 0) and the flow is directed to the outlet manifold. The bypass cleaning volume is ~800 mL. After a cycle of particle accumulation and imaging, the motor driven sampler rotator opens to one of four

sample bottle positions (1 – 4, Fig. A2) and the suction action of the pump draws water and particles from the imaging stage into the selected 250 mL conical clarified polypropylene centrifuge tube (Thermo Scientific, Nunc). A total of ~400 mL of water is drawn through the sampling system during each regular cleaning cycle and represents about a 30% of the volume of the collection funnel (~1460 cm$^3$). All particle transfers from a dive are directed to the same bottle (diamond points in Fig. A1). Particles are retained in the bottle by a 14 cm diameter circle of 51 μm polyester 33% open area mesh (SEFAR 07-51/33)

wrapped and secured using silicone o-rings around the outlet structure within the bottle (Fig. 2c). The area of perforated outflow cylinder was ~30 cm$^2$; however, when the circular mesh was secured to the top of the outlet cylinder by an o-ring, the pleated mesh area exposed to flow was ~130 cm$^2$ (Fig. 2c).

The flow from imaging stage to bottle is constricted by the six 3 mm diameter openings that surround the sample stage. Loosely aggregated material is likely distorted or broken up into smaller pieces while being transferred. For cohesive particles (such as

Siphonophores) and rigid particles (such as some Pteropod shells), the upper size limit is 3 mm. Though nothing was caught in the CFE-Cals during this study, there was one case where a larval crab was caught in one of the profiling CFEs and was not able to fit through the 3 mm diameter opening. This had to be removed after the CFE was retrieved.

### 2.3 Sampler Materials

Little is known about water absorption properties, dimensional stability, and chemical reactivity and contamination potential

of the 3D printing resins as most are proprietary. The majority of sampler parts were fabricated using the Connex3 from FullCure 720 resin (Fig. A2) and some of the particle isolation assemblies were printed in both FullCure 720 and VeroWhite RGD35 resins. The Connex3 is a fused deposition modeling (FDM) printer which builds parts layer by layer. We fabricated three additional particle isolation assemblies from amber Cyanate Ester, black rigid Polyurethane and black Polylactic (PLA) resins on the Carbon printer; the process uses photopolymerization to form a solid piece as material is drawn from liquid resin.

After parts were printed and support material removed, the parts were rinsed with deionized water and then leached in a 1.2 M HCl solution for 16 hours at room temperature. All remained stable to this treatment. Dimensional tests before and after sea trials showed that dimensions of the sampler body (Fig. A2) printed with FullCure 720 remained stable to within 0.06% of design dimensions.

## 2.4 Field Procedures

Prior to each deployment, the CFEs sample stage and related glass surfaces were cleaned to remove any remaining material collected during the previous deployment. Areas between glass layers were flooded with water to prevent air bubbles being trapped. Each CFE-Cal was outfitted with four clean sample collection tubes and filled with 0.4 μm filtered seawater. On recovery of the CFE-Cals, the sample bottles (Fig. 2d) were either immediately removed from the sampler and filtered or placed in a fridge at 10 °C to minimize sample degradation; in the latter case, samples were processed within 3 hours of collection.

All sample processing and manipulation took place in a laminar flow bench at sea. Each sample was decanted into an open filter funnel loaded with either 47 mm diameter Whatman Quartz Fiber (QMA, pore size ~1.2 μm) or Supor (pore size 0.4 μm) filters; transfer took place with filters under mild suction with the aim of evenly covering the filter surface (Fig. A3). Each sample tube and associated 51 μm mesh were further rinsed three times with ~ 5 mL of 0.4 μm filtered seawater to ensure quantitative transfer of particles. After filtration, the samples were quickly misted with ~3 mL of deionized water (DI) to reduce residual sea salt while still under suction. Samples were then placed in Gelman Petri slides and photographed wet under LED ring light illumination using a 20 Mpixel Sony RX100 V camera (pixel resolution of 19 μm), dried at 50 °C for 24 hours, and photographed again under the same lighting conditions in a laminar clean air bench. Dried samples were then stored in covered petri slides until analyzed in the laboratory. Prior to use, the QMA filters were placed in a muffle furnace at 450 °C overnight to reduce carbon blanks. Both the QMA (after combustion) and the Supor filters were leached in a 1.2 M HCl solution for 24 hours at room temperature and rinsed with deionized water and air dried in a class 100 laminar flow bench prior to use.

## 2.5 Laboratory Procedures

### 2.5.1 Carbon and Nitrogen Analysis

Briefly, half of each QMA filter was placed in a desiccator and exposed to HCl fumes (from 12 M HCL) for 24 hours to remove any carbonate carbon (Bishop et al. 1978) and then dried at 30°C for 36 hours and subsampled 6 to 8 times using a 3 mm diameter biopsy punch yielding ~1/16th of the whole sample. These were loaded into tin capsules and analyzed for total organic carbon and nitrogen using a Thermo Quest EA2500 Elemental Analyzer at Oregon State University according to Holser et al. (2011). A total of 27 unique cruise samples and process blanks (with 6 replicates), 5 unused QMA filters, and analytical blanks (empty tin capsules) were run. Process blanks were samples where no particles were directed to sample tube during deployment and processed as other samples. The other half of the sample was preserved for ICP-MS analysis.

Corrected POC was calculated following Eq (1):

$$POC_{corrected} = POC_{measured} - POC_{process\ blank}$$

The sample POC error was calculated following Eq (2):

$$POC_{error} = \sqrt{(process\ blank\ s.d.)^2 + (sample\ RSD \times POC_{corrected})^2}$$

Nitrogen and phosphorous were calculated the same way, replacing POC with PN and PP.

### 2.5.2 ICP-MS Phosphorous Analysis

Samples on both Supor and QMA filters were analyzed using a Thermo Fisher Element II XR Inductively Coupled Plasma Mass Specrometer (ICP-MS) at the UC Santa Cruz Marine Analytical Laboratory following Bishop et al. (2012). Half of each 47 mm filter was leached in 10 mL of a 0.6 M HCl solution at 60°C for 16 hours. The leach solution was then diluted with 18.2 mOhm-cm Milli-Q DI water to 50 grams; 1 mL of the diluted solution was then further diluted with 3 mL of 0.12 M HCl and spiked with 0.2 mL of 25 ppb In. Standards were prepared in the same acid matrix.

### 3 Results and Discussion

### 3.1 Samples Collected

Samples were collected from four productivity regimes and environmental conditions yielding a diverse array of particle sizes and classes (Fig. 3a-e). The flux rates between locations also varied widely. At location 1, flux was at times dominated by 1 mm diameter, 5-10 mm long anchovy pellets similar to those described by Saba and Steinberg (2012) with 95% of VA flux (average ~40 mATN-cm$^2$ cm$^{-2}$ d$^{-1}$) being carried by particles > 1.5 mm in size. In contrast, at location 2, numerous small diameter (200-300 µm) olive green ovoid pellets dominated imagery and accounted for ~ 50% of the ~15 mATN-cm$^2$ cm$^{-2}$ d$^{-1}$ VA flux. Location 3, in transitional waters near the filament edge, had a VA flux of ~2.3 mATN-cm$^2$ cm$^{-2}$ d$^{-1}$ and ~65% of the flux carried by aggregates larger than 1.5 mm. At Location 4, in the most extended part of the filament, VA flux was ~ 22 mATN-cm$^2$ cm$^{-2}$ d$^{-1}$, and 94% of the flux was carried by aggregates >1.5 mm in diameter.

The CFE-Cals were new and there were initial malfunctions (i.e. instrument not diving to depth, not stabilizing at depth, or sampler not switching target bottles correctly) which were mostly resolved during the first half of the expedition. In all, The CFE-Cals were deployed 15 times over the course of the June 2017 CCE-LTER study, the CFE was outfitted with 4 sample bottles for each deployment. For each deployment, depending on how much time was available, the CFE-Cals performed 3 to 4 dives. Of these 60 possible sample collections, 10 were not useable due to a CFE-Cal malfunction, and one was not useable due to a swimming organism. Almost all these malfunctions occurred at the beginning of the cruise. Table 1 details all these points as well as noting sampling times, depths and filter type. The CFE-Cal instruments were built using the new SOLO2 floats whereas the original CFEs were built using SOLO1 floats. We found that the concave bladder housing of the new SOLO2 float design trapped air and made it more difficult for the CFE to attain its target depth. Once we realized this issue, before each deployment, the bottom of the float was flushed with water prior to each deployment and care was made to launch the

float horizontally to prevent this. As all instrument malfunctions were identified and resolved, future deployments will be far more robust.

A total of 15 QMA samples were analysed for POC and PN; these 15 QMA samples and 19 Supor samples were analysed for PP. Samples ranged from 0.0267 to 0.1570 mmol C/filter (average ± sd: 0.0760 ± 0.0362) and 0.0029 to 0.0155 mmol N/filter (average ± sd: 0.0065 ± 0.0034). Phosphorous in samples ranged 40-fold from $3.9\times10^{-5}$ to $1.5\times10^{-3}$ mmol P/filter (average ± sd: $2.0\times10^{-4} \pm 2.4\times10^{-4}$). Results are shown in table S1.

Process blanks were subtracted from the sample values as shown in equation 1. As there were only 5 QMA process blanks, an average of the five was used to blank correct POC and PN. This drove one POC and one PN sample from Location 3 negative, though not negative within error. Fluxes at location 3 were very low - an order of magnitude lower than samples collected in other regions. Process blanks contained 0.032 ± 0.008 mmol C/filter and 0.003 ± 0.0003 mmol N/filter. Unused QMA blanks were 0.0037 ± 0.0008 mmol C/filter and were below the detection limit for nitrogen; only 12% of carbon in the process blanks came from the blank filter. Nearly 90% of the process blank carbon is therefore not from the blank filter, but stems from either accidental collection of particles during deployment or contamination during processing. Particles may be accidentally collected by entering a sample bottle while the sampler is turning and the selector briefly passes the blank bottle inlet.
As there were at least two process blanks per location for PP, blanks could be location specific. The process blanks were $8.9\times10^{-5}$, $5.0\times10^{-5}$, $1.9\times10^{-5}$ and $5.0\times10^{-5}$ mmol P/filter for location 1, 2, 3 and 4 respectively.

Replicate analysis of 4 samples gave an average RSD of 0.14 and 0.07 for C and N respectively. Punched subsamples are collected evenly distributed across the filter, but inevitably as there are discrete particles on the filter, there is some heterogeneity between the sub-samples (Figure A5). The RSD of replicate analysis we assume is attributed to this sample heterogeneity and can be applied to all samples. The RSD for replicate analyses of process blanks was 0.18 and 0.12 for C and N respectively.

### 3.2 Transfer Efficiency

To validate the efficiency of transfer of particles imaged to sample bottles, ovoid pellets (200-300 µm) were manually counted (Fig. 4) in both the CFE's OSR images and of photographs of filters of material sampled at location 2. CFE-Cal2 collected close to the same number of particles in the sampler as were imaged (on average, there was less than a 9% difference between particles imaged and particles filtered, particles were not exclusively higher in one or the other). CFE-Cal4, however, collected 1.45 times more ovoid pellets in the sampler than were imaged. The sampler uses an optical encoder to sense a home position from which it advances to select specific bottles. Software is programmed with a time out in the case that the "home" position cannot be found to prevent continuous operation and depletion of the CFE batteries. During pre-cruise tests of the CFE-Cal no

positioning errors were registered. However, these tests were done in a lab, with room temperature fresh water and not in 10°C seawater under pressure. It is clear that the sampler for CFE Cal4 stopped short of home position, and this likely led to the over transfer of particles that were not imaged. We recognized this issue during deployment operations at cycle 3 and after the time-out limit was adjusted; this problem was not encountered again. The known sampler positioning issues at this time may

have led to transfer of pellets to bottles from times (such as during float ascent to the surface) when particles were not imaged by CFE-Cal4. To correct for this, the POC, PN and PP numbers for CFE-Cal4 at location 1, 2 and 3 were divided by 1.45.

Bishop et al., (2012) investigated the effect of filtration rate on aggregate retention during large volume in-situ filtration sampling and found that aggregates were broken up when the flow velocity through 51 μm mesh exceeded 1 cm s$^{-1}$ over a

four-hour sampling time. During CFE-Cal stage cleaning, the sample transfer pump is operated for two cycles of ten seconds at a flow rate of ~20 mL s$^{-1}$. The mesh area on the outflow from the sample bottle is approximately 130 cm$^2$. We thus calculate the flow speed through the mesh to be ~0.15 cm sec$^{-1}$, 15% of the threshold speed recommendation by Bishop et al. (2012). Although intact large aggregates were not seen on the sample filters (compare Fig. A3d vs. Fig. 3d), given our limited sample transfer time (< 2 minutes) and low flow velocity, we believe that our transfer efficiency for the particles comprising the

loosely aggregated material is similar to that for the more robust pellets.

### 3.3 Calibration Results

Figure 5 shows cumulative VA regressed against sample POC, PN and PP (data in Table S1). All of our results are forced through zero as both VA and elemental values are blank controlled. Regressions results yielded slopes and R$^2$ values and number of samples (in parentheses) of 1.01 x 10$^4$ mATN-cm$^2$: mmol POC (r$^2$=0.86, n=12, p<0.001) and 1.01 x 10$^5$ mATN-

cm$^2$: mmol PN (r$^2$=0.86, n=15, p<0.001). Three of 15 samples had C/N ratios above 20 and were not used in the regression for POC as these numbers are not typical of sinking particles (e.g. Bishop et al., 1977, Lamborg et al., 2008, Stukel et al., 2013). Stukel et al. (2013) reported trap POC/PN mole ratios ranging from 5-14 (average, 9.6) at 100 m in the same upwelling regime we have sampled; Lamborg et al. (2008) reported POC/PN ratios ranging from 7.7 to 8.5 in productive waters of the Oyashio and Oligotrophic waters of the North Pacific Gyre. The molar ratio of 10 C/N from our regression

slopes is in line with Stukel et al. (2013).

The high C/N values of excluded samples may have been due to contamination by residual material used as a scaffold to build the 3D printed parts; in one case, a 1 mm sized aggregate of such material was found on our filters. New lighting systems were rapidly prototyped just prior to cruise. The scaffold material, Stratasys' OBJET Support SUP706 is made of

1,2-Propylene glycol and Polyethylene glycol, Methanone, (1-hydroxycyclohexyl) phenyl-both of which contain carbon but not nitrogen (SUP706 SDS https://store.stratasys.com/medias). The material also contains an unspecified acrylic. As the 3D printed material contains no nitrogen, C/N values would be elevated if they were contaminated. Including the three high C/N

ratio points in the VA:POC regression reduces the slope to $0.86 \times 10^4$ with an $R^2$ of 0.64 (n=15, p<0.001); we report this even though we do not believe this to be representative of natural particulates.

The data also demonstrate that there is no obvious difference for VA:PN or VA: POC for samples collected from Locations 1 and 4 (Fig. 1, Fig. 4) where aggregates > 1.5 mm in size accounted for 95% of the flux compared to Locations 2 and 3 where smaller material contributed 50 and 30% of the flux, respectively.

The relationship for VA: PP was scattered with a slope of $1.53 \times 10^6$ mATN-cm$^2$: mmol PP ($R^2$=-0.07, negative $R^2$ values denote results worse than horizontal fit). The sample with anchovy fecal pellets had a POC/PP ratio of 90 mol/mol, far higher than all other samples (~300 mol/mol). The fact that PP had zero correlation with VA is consistent with the strong loss of P relative to carbon and nitrogen as large aggregates sink (e.g. Bishop, 1977; Lam et al., 2007). Scanning electron microscopy showed that the anchovy fecal pellets were stuffed with diatoms and as they are larger and sink at a much faster rate (up to 500m d$^{-1}$), it follows that this sample should have a higher PP content as there is less time for microbial degradation and remineralization. and when this point is arbitrarily removed, the relationship of VA: PP for non-Anchovy aggregates improved $3.23 \times 10^6$ mATN-cm$^2$ : mmol PP ($R^2$=0.41). Though the ultimate goal is to allow an estimation of biogeochemical fluxes based on image analysis, because of the known lability of PP relative to POC or PN in sinking particles, we conclude that PP cannot be predicted from VA.

As mentioned earlier, we found that CFE-Cal4 collected 1.45 times more ovoid pellets on the filters than were imaged due to a sampling issue and that we therefore divided the CFE-Cal4 POC and PN samples for location 1, 2 and 3 by this empirically derived factor. This affected 6 samples in the POC and PN regressions (see table S1). We note that if instead of applying this empirical factor, these samples are removed from the regression, the VA: POC and VA:PN slopes both change less than 5%. The slopes (number of samples and $R^2$ in parenthesis) change to $1.06 \times 10^4$ (n=8, $R^2$=0.93) and $1.04 \times 10^5$ (n=9, $R^2$=0.91). Using these data which have been corrected using the empirical factor therefore affects the overall regression very little.

### 3.4 Comparison to previous studies

Two autonomous flux monitoring systems, the CFE (Bishop et al., 2016) and the OST (Estapa et al., 2017), have now been calibrated to relate the attenuance flux to the flux of particulate organic carbon. This study expands upon Estapa et al. (2017) as samples from a wide range of environments and with a far greater range of aggregate size distributions have been collected. The highest POC flux collected in Estapa et al.'s (2017) calibration was under 2 mmol C m$^{-2}$ d$^{-1}$. The flux environments sampled in our study ranged from <2 to 40 mmol C m$^{-2}$ d$^{-1}$.

Figure 6 compares the relationship between VA flux and carbon flux from this study (data for regression in Table S1) vs. data from Estapa et al. (2017). When converted to compatible units, the slope for Estapa's VA flux (mATN-cm$^2$ cm$^{-2}$ d$^{-1}$) versus POC flux (mmol C m$^{-2}$ d$^{-1}$) is 1.50 (allowing for an intercept) and 2.19 (forced through zero) and while our slope is 1.03 (forced through zero). Estapa et al. (2017) calculates attenuance by taking the natural log of transmittance and observations are reported in units of ATN m$^2$ m$^{-2}$ d$^{-1}$. Our data are the log$_{10}$ of transmittance as documented in Bishop et al. 2016 and reported in units of mATN cm$^2$ cm$^{-2}$ d$^{-1}$. Therefore, Estapa's published data has been divided by 2.303 to convert the natural log attenuance to log$_{10}$ attenuance and multiplied by 1000 to scale to mATN units. The dimensional data do not require scaling. Our observations were for depths near 150 meters and it is unknown if there is a depth dependence to calibration factors. We note that Estapa et al. (2017) combined samples from ~150, 300 and 500 meters in her regression. This said, the slopes of our two datasets differ by a factor of 1.5 to 2. In our data, attenuance of particles in the red image plane is 6% lower than in green, thus wavelength of analysis is a minor factor explaining the differences. Given the large range in particle size distributions and classes (e.g samples dominated by dense Anchovy pellets vs. 2-3 mm size amorphous semi-transparent aggregates, we can rule out particle size effects. Beam geometry and the other factors underlying our different methodologies likely explain the differences found.

As noted earlier, Bishop et al. (2016) estimated the factor for conversion of POC flux (C mmol m$^{-2}$ d$^{-1}$) to VA flux (mATN-cm$^2$ cm$^{-2}$ d$^{-1}$) using two previously published values of carbon: particle volume as there were not calibration samples available. The two methods are described briefly. In method one, Bishop et al. (2016) calculated an aggregate volume of 0.113 cm$^3$ for particles >800 μm (which accounted for 97% of VA) in a series of 5 images. An aggregate density of 0.087 g cm$^3$ and 60% organic matter of dry weight was assumed based on Bishop et al., (1978). To convert the organic matter weight to carbon, a conversion factor of 1.88 was used from Hedges et al. (2002). Using these values yielded an estimated flux of 183 mmol C m$^{-2}$ d$^{-1}$, compared to a VA flux of 66.2 mATN-cm$^2$ cm$^{-2}$ d$^{-1}$, which gives a POC flux (C mmol m$^{-2}$ d$^{-1}$) to VA flux (mATN-cm$^2$ cm$^{-2}$ d$^{-1}$) of 2.8 (183: 66.2× 0.97=2.8). For comparison, POC was also estimated following the methods of Alldredge (1998). Alldredge (1998) derived a regression formula relating the equivalent spherical volume (ESV) and POC content of particles:

$$POC_{\mu g} = 0.99 \times ESV^{0.52}$$

The Alldredge (1998) relationship has previously been used to estimate carbon content from aggregates in gel trap imagery (Ebersbach and Trull, 2008; Ebersbach et al., 2011) and aggregates collected then imaged using Marine Snow Catchers (Riley et al., 2012; Baker et al., 2017). In the Bishop et al. (2016) study, the POC density of aggregates was 0.028 g C cm$^{-3}$ following method 1, and 0.0002 g C cm$^{-3}$ using the Alldredge (1998) formula. The two order of magnitude difference was likely due to the fact that the relationship between aggregate volume to POC for Alldredge (1998) was based on aggregates collected between 10 and 20 m, photographed in-situ in a plane parallel to their sinking direction, whereas the Bishop et al. (2016) method was based on published values of large particles collected using large volume filtration between 100 and 400 m depth. The conversion factor for POC:VAF for Alldredge was 17 times lower than the Bishop et al. 2016 factor of 2.8 (Bishop et al.,

2016). Analysis of directly imaged and sampled material in this study yielded a slope of 1.03 for VAF:POC, which is about 3 times higher than estimated using Bishop et al., (1978) but 6 times lower than inferred from Alldredge et al. (1998) (Figure 6). It is not surprising our results are not consistent with Alldredge (1998) as our samples were collected more than 100m deeper and Alldredge's volumes were calculated based on images taken in-situ parallel to the aggregate sinking direction whereas the

CFE images are collected looking upwards after the aggregates have settled onto the glass stage. Though the settling motion is gentle and does not break apart the aggregate, there is very likely some compaction as the particle settles onto the stage.

Bishop et al. (2016) reported CFE attenuance fluxes averaging 66.2 mATN-cm$^2$ cm$^{-2}$ d$^{-1}$ at 150 m in the Santa Cruz Basin in January 2013 and estimated a POC flux of 190 mmol C m$^{-2}$ d$^{-1}$, about 8 times higher than the highest previously measured flux

from surface-tethered sediment traps deployed over a 3-year period at 100 and 200 meters in nearby waters (Thunell, 1998; August 1993 to September 1996). Converting the 66.2 mATN-cm$^2$ cm$^{-2}$ d$^{-1}$ attenuance flux to POC flux using our new calibration yields 64.3 mmol C m$^{-2}$ d$^{-1}$, a value which is still three times higher than the highest previously measured flux (Thunell, 1998). In short, the likely discrimination of surface tethered baffled sediment traps against the collection of >1 mm sized particles remains an issue in biologically dynamic regimes dominated by large aggregates.

## 4 Conclusions and future development

We have presented the initial calibration of the CFE optical proxy for particulate carbon and nitrogen using the newly developed CFE-Cal instruments. The development of a sampling system for the Carbon Flux Explorer has overcome a major

barrier to the calibration of our attenuance proxy for organic matter export. The calibration of volume attenuance flux (VAF) against organic carbon and nitrogen flux in this study represents an important step forward in the development of autonomous optical flux measurements. Our regression results yield well-correlated calibrations for POC and PN (POC R$^2$ = 0.86, n=12, p<0.001 and PN R$^2$ = 0.86, n=15, p<0.001) that apply over a wide range of environments, including high flux environments in recently upwelled water as well as low flux off-shore transitional waters. Phosphorus was shown to be poorly correlated,

consistent with the highly labile nature of this element relative to either C or N. Our results give us confidence that images collected by the CFE can be used to calculate the fluxes of carbon and nitrogen. We find less than a two-fold difference in the POC flux vs. Volume Attenuance flux regression slope from Estapa et al. (2017). This is remarkable given the strongly different environments, methodology, and means by which fluxes were sampled. Both these studies reinforce the theory that light attenuation can be used as a proxy for POC and in our case PN flux.

Optimization of CFE-Cal sample return, performance validation, and simplification of recovery logistics during CCE-LTER required that all samples in this dataset to be collected near 150 m. These calibration samples were all collected off the coast

of California, during the month of June near 150 meters. To make the calibration more robust and determine whether the calibration relationships derived here are widely applicable, it is essential to extend these results to greater depths, and to different oceanic regions, environments and ecosystem structures. Intercalibration of the CFE attenuance measurements with other autonomous systems should be pursued.

Results presented above demonstrate that the magnitude of flux and of food web processes responsible for flux can vary strongly over relatively small spatial and temporal scales in dynamic coastal waters. Thus, the use of high frequency autonomous observations will significantly better inform food web and carbon export simulations. Our promising initial calibration of VAF in terms of POC and PN justifies further development of instruments which optically measure POC such

as the Carbon Flux Explorer.

**Acknowledgements**

We would like to thank Mark Ohman (chief scientist), members of the science party, and captain and crew of the R/V Revelle for facilitating CFE-Cal deployments during the June 2 – July 1 2017 California Current Ecosystem Long Term Ecological Research process study. Lee-Huang Chen (UC Berkeley, Engineering) contributed at critical stages of this project. We thank Alejandro Morales (LBNL), Christopher Parsell and the Jacobs Institute for Design Innovation (UC Berkeley), Christopher Myers (UC Berkeley), Mike McLune (SIO), Phoebe Lam and Rob Franks (UC Santa Cruz). We'd also like to thank the many

UC Berkeley undergraduates who have worked with us at sea and in the laboratory, in particular, Casey Fritz, Beth Connors, Xiao Fu, Sylvia Targ, Jessica Kendall-Bar, and William Kumler. US National Science Foundation grant OCE 0936143 supported Carbon Flux Explorer development; OCE 1528696 supported the development and at sea work with the CFE-Cal systems. The CCE-LTER is supported by US NSF grant OCE 16-37632.

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

| Study Loc. | CFE | Latitude (°N) | Longitude (°W) | Dive | Bottle | Filter | UTC Day Start | UTC Day End | Hours | Depth (m) | Depth stdev | Sample Notes |
|---|---|---|---|---|---|---|---|---|---|---|---|---|
| 1 | 2 | 35.0739 | 121.1281 | 40 | 1 | Supor | 160.064 | 160.270 | 4.944 | 0.2 | | CFE-Cal did not dive |
| 1 | 2 | | | 41 | 2 | QMA | 160.333 | 160.500 | 4.008 | 70.0 | 12.3 | |
| 1 | 2 | | | 42 | 3 | Supor | 160.623 | 160.791 | 4.032 | 119.4 | 7.8 | not analyzed |
| 1 | 2 | 34.9978 | 121.1650 | 43 | 4 | QMA* | 160.849 | 160.851 | 0.048 | 186.0 | | |
| | | | | | | | | | | | | |
| 1 | 4 | 35.0885 | 121.1293 | 40 | 1 | QMA | 160.076 | | | | | selector failure |
| 1 | 4 | | | 41 | 2 | Supor | 160.297 | | | | | selector failure |
| 1 | 4 | | | 42 | 3 | QMA | 160.422 | | | | | selector failure |
| 1 | 4 | 35.0341 | 121.1862 | 43 | 4 | Supor | 160.732 | | | | | selector failure |
| | | | | | | | | | | | | |
| 1 | 2 | 34.9396 | 121.2031 | 50 | 1 | QMA | 162.091 | 162.294 | 4.872 | 131.7 | 10.9 | |
| 1 | 2 | | | 51 | 2 | Supor | 162.424 | 162.536 | 2.688 | 31.8 | 4.4 | not analyzed |
| 1 | 2 | | | | 3 | QMA* | | | | | | |
| 1 | 2 | 34.8962 | 121.2032 | | 4 | Supor* | | | | | | |
| | | | | | | | | | | | | |
| 1 | 4 | 34.9348 | 121.1946 | 50 | 1 | Supor | 162.075 | 162.280 | 4.908 | 189.5 | 7.7 | oil under sample stage |
| 1 | 4 | | | 51 | 2 | QMA | 162.410 | 162.412 | 0.048 | 286.2 | | depth unstable |
| 1 | 4 | | | 52 | 3 | Supor | 162.549 | 162.551 | 0.048 | | | depth unstable |
| 1 | 4 | 34.8997 | 121.2165 | | 4 | QMA* | | | | | | |
| | | | | | | | | | | | | |
| 2 | 2 | 34.7771 | 122.0572 | 60 | 1 | Supor | 165.047 | 165.264 | 5.208 | 142.9 | 3.8 | |
| 2 | 2 | | | 61 | 2 | Supor | 165.406 | 165.574 | 4.032 | 112.8 | 4.3 | |
| 2 | 2 | | | 62 | 3 | Supor | 165.716 | 165.883 | 4.008 | 97.7 | 4.1 | |
| 2 | 2 | 34.8651 | 122.3355 | | 4 | QMA | 166.024 | 166.026 | 0.048 | | | surfaced open at position 4 |
| | | | | | | | | | | | | |
| 2 | 4 | 34.7742 | 122.0587 | 60 | 1 | QMA | 165.060 | 165.278 | 5.232 | 160.9 | 4.2 | C:N >20 |
| 2 | 4 | | | 61 | 2 | Supor | 165.430 | 165.597 | 4.008 | 153.1 | 5.7 | not analyzed |
| 2 | 4 | | | 62 | 3 | QMA | 165.739 | 165.900 | 3.864 | 150.2 | 3.0 | |
| 2 | 4 | 34.8825 | 122.3499 | | 4 | Supor* | | | | | | |
| | | | | | | | | | | | | |
| 2 | 2 | 34.7098 | 122.3004 | 70 | 1 | Supor | 166.659 | 166.882 | 5.352 | 159.2 | 5.1 | |
| 2 | 2 | | | 71 | 2 | QMA | 167.034 | 167.202 | 4.032 | 146.2 | 5.8 | |

| | | | | | | | | | | | | |
|---|---|---|---|---|---|---|---|---|---|---|---|---|
| 2 | 2 | | | 72 | 3 | Supor | 167.350 | 167.517 | 4.008 | 147.8 | 3.9 | |
| 2 | 2 | 34.6771 | 122.4122 | | 4 | QMA* | | | | | | |
| | | | | | | | | | | | | |
| 2 | 4 | 34.7091 | 122.2998 | 70 | 1 | QMA | 166.673 | 166.897 | 5.376 | 164.2 | 10.3 | |
| 2 | 4 | | | 71 | 2 | Supor | 167.044 | 167.211 | 4.008 | 157.6 | 3.4 | |
| 2 | 4 | | | 72 | 3 | QMA | 167.364 | 167.531 | 4.008 | 151.5 | 2.9 | C:N >20 |
| 2 | 4 | 34.6829 | 122.4185 | | 4 | Supor* | | | | | | |
| | | | | | | | | | | | | |
| 3 | 2 | 34.2275 | 123.1480 | 80 | 1 | QMA | 170.192 | 170.368 | 4.224 | 141.5 | 6.7 | |
| 3 | 2 | | | 81 | 2 | Supor | 170.472 | 170.639 | 4.008 | 131.4 | 3.8 | |
| 3 | 2 | | | 82 | 3 | QMA | 170.740 | 170.879 | 3.336 | 143.7 | 6.7 | C:N >20 |
| 3 | 2 | 34.1717 | 123.0758 | | 4 | Supor* | | | | | | |
| | | | | | | | | | | | | |
| 3 | 4 | 34.1129 | 122.9885 | 90 | 1 | QMA | 171.205 | 171.414 | 5.016 | 173.4 | 3.3 | |
| 3 | 4 | | | 91 | 2 | Supor | 171.553 | 171.721 | 4.032 | 160.9 | 0.1 | |
| 3 | 4 | | | 92 | 3 | QMA | 171.860 | 171.903 | 1.032 | 148.8 | 0.8 | |
| 3 | 4 | 34.0749 | 122.8673 | | 4 | Supor* | | | | | | |
| | | | | | | | | | | | | |
| 3 | 2 | 34.1086 | 122.9823 | 90 | 1 | Supor | 171.190 | 171.369 | 4.296 | 126.9 | 4.8 | |
| 3 | 2 | | | 91 | 2 | QMA | 171.468 | 171.636 | 4.032 | 159.7 | 5.7 | |
| 3 | 2 | | | 92 | 3 | Supor | 171.737 | 171.904 | 4.008 | 154.7 | 3.6 | |
| 3 | 2 | 34.0714 | 122.8552 | | 4 | QMA* | | | | | | |
| | | | | | | | | | | | | |
| 4 | 4 | 34.4070 | 123.0958 | 100 | 1 | Supor | 174.180 | 174.369 | 4.536 | 190.6 | 5.7 | |
| 4 | 4 | | | 101 | 2 | Supor | 174.489 | 174.657 | 4.032 | 117.3 | 3.3 | |
| 4 | 4 | | | 102 | 3 | Supor | 174.767 | 174.899 | 3.168 | 135.0 | 3.5 | |
| 4 | 4 | 34.4174 | 123.0535 | | 4 | Supor* | | | | | | |
| | | | | | | | | | | | | |
| 4 | 2 | 34.4032 | 123.0964 | 100 | 1 | QMA | 174.294 | 174.354 | 1.440 | 165.8 | 132.5 | depth unstable |
| 4 | 2 | | | 101 | 2 | Supor | 174.479 | 174.646 | 4.008 | 139.9 | 3.0 | |
| 4 | 2 | | | 102 | 3 | QMA | 174.742 | 174.903 | 3.864 | 129.5 | | |
| 4 | 2 | 34.4216 | 123.0310 | | 4 | Supor* | | | | | | |
| | | | | | | | | | | | | |
| 4 | 4 | 34.4221 | 123.0133 | 110 | 1 | QMA | 175.187 | 175.487 | 7.200 | 164.0 | 5.4 | |
| 4 | 4 | | | 111 | 2 | Supor | 175.599 | 175.878 | 6.696 | 101.7 | 3.7 | |
| 4 | 4 | | | 112 | 3 | QMA | 175.989 | 176.267 | 6.672 | 158.6 | 3.9 | |

| | | | | | | | | | | | | |
|---|---|---|---|---|---|---|---|---|---|---|---|---|
| 4 | 4 | 34.4449 | 123.0205 | 113 | 4 | Supor | 176.396 | 176.496 | 2.400 | 119.8 | 1.8 | |
| 4 | 2 | 34.4218 | 123.0168 | 110 | 1 | Supor | 175.173 | 175.469 | 7.104 | 162.8 | 5.5 | |
| 4 | 2 | | | 111 | 2 | QMA | 175.582 | 175.859 | 6.648 | 159.5 | 3.4 | swimming siphonophore |
| 4 | 2 | | | 112 | 3 | Supor | 175.965 | 176.242 | 6.648 | 156.9 | 3.0 | |
| 4 | 2 | 34.4335 | 123.1008 | 113 | 4 | Supor | 176.350 | 176.516 | 3.984 | 153.1 | 2.3 | |

**Table 1: CFE-Cal 2017 deployments during the California Current Ecosystem Long Term Ecological Research process study**

**This table notes the location number, CFE number, longitude and latitude position, dive number, bottle number, filter type, sampling interval and depth interval of sampling. Filters marked with the \* symbol are process blanks. The note column indicates any sample notes, including any instrument malfunctions.**

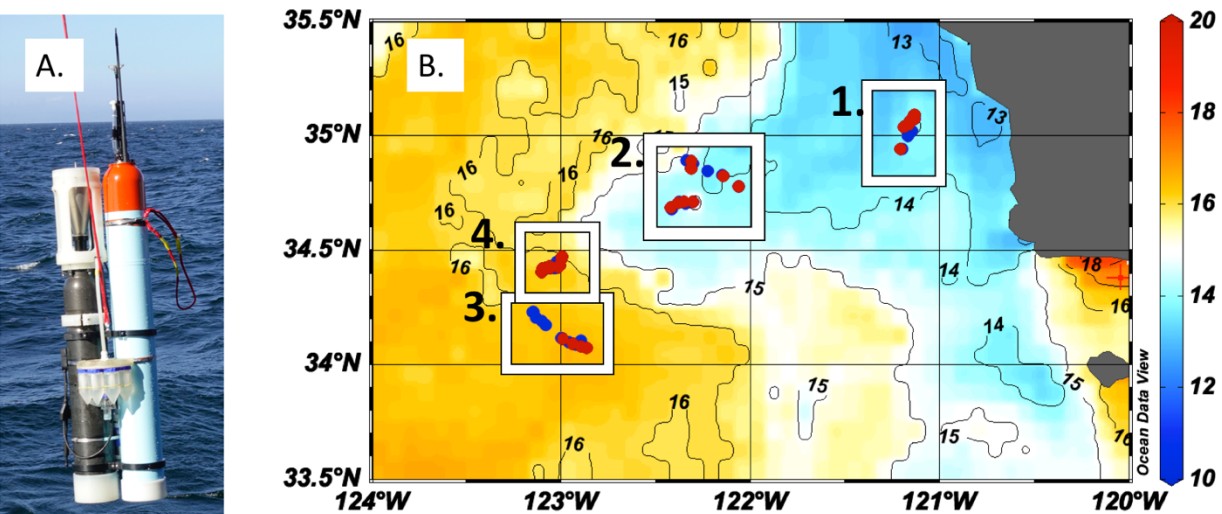

Figure 1: (A) CFE-Cal during deployment from R/V Revelle in 2017. The sampling system for particles is interfaced between the Optical Sedimentation Recorder (left) and SOLO float (right). (B) Map of CFE-Cal deployment and drift locations overlaying map of sea surface temperature (°C) for June 10-17 2017 from NASA Ocean Color Aqua Modis 4km resolution (https://oceancolor.gsfc.nasa.gov/). Blue dots within location boxes represent CFE-Cal 002 and red dots represent CFE-Cal 004 positions.

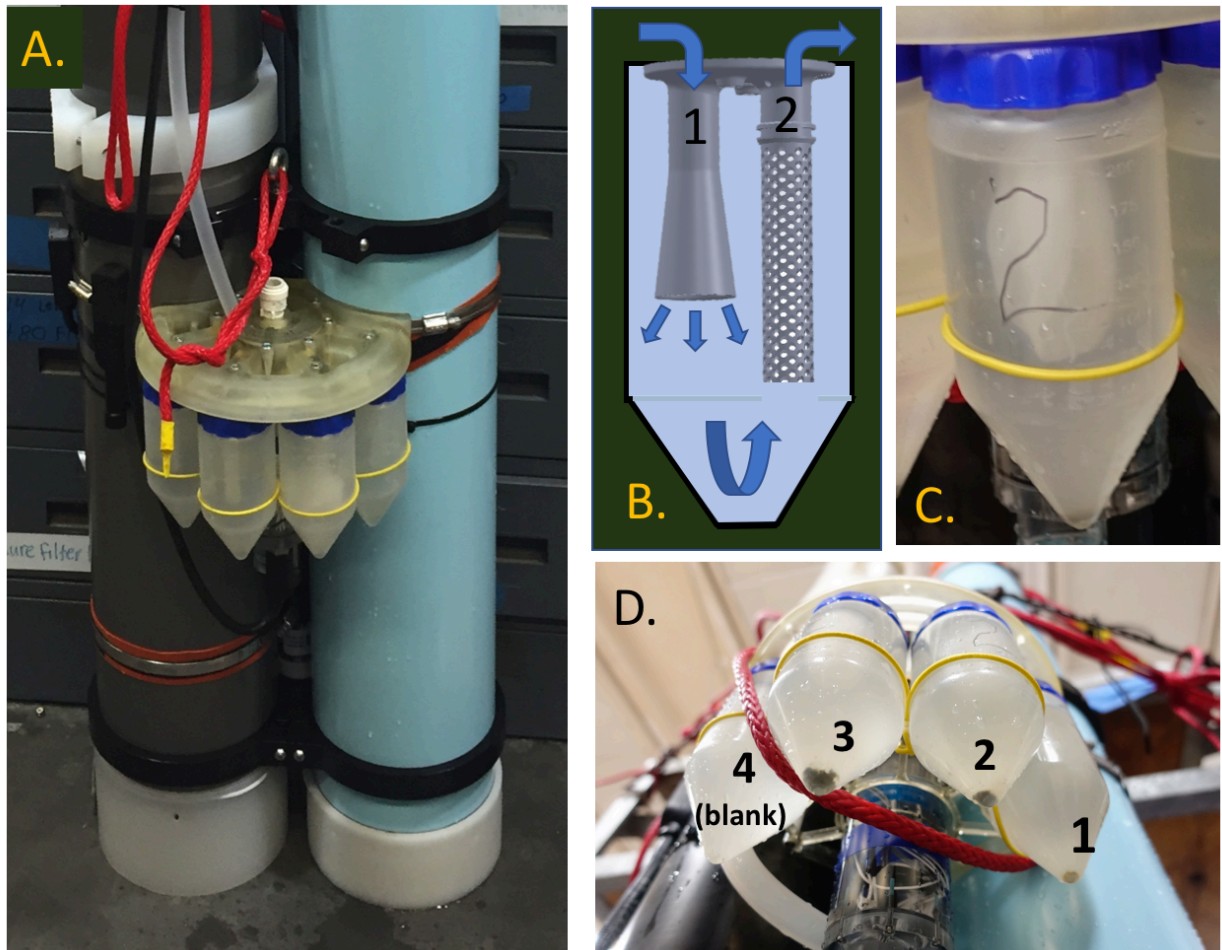

Figure 2: (A) Sampler on CFE-Cal. Suction action of a pump draws water and particles down a poly tube to the sampler (shown disconnected). (B) Detail of particle retention system within sample bottles. Inlet is cone shaped to decelerate incoming flow. Outlet is formed to accommodate 51 μm mesh which is retained by two o-rings at the top. (C) Closeup of bottle with Mesh filter in place; Filter area is ~130 cm². (D) CFE-Cal recovery after 24-hour deployment showing collected samples. Bottle 2 is shown in C. In this case, bottle 4 was a blank (i.e. no particles directed to it).

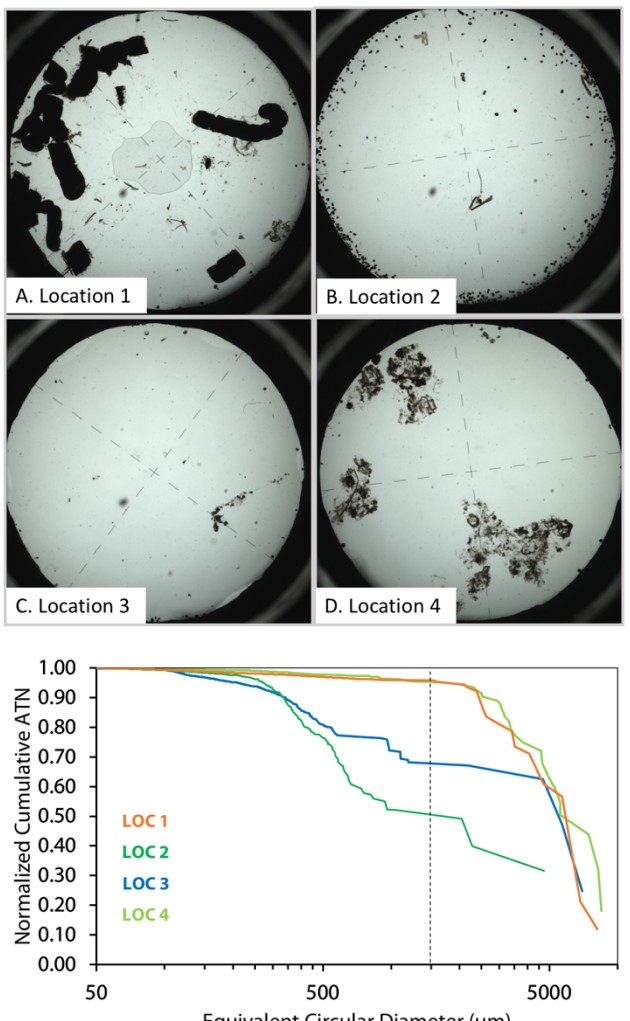

Figure 3: Representative images from four locations. The particle size classes present varied widely at the four different locations. (A) In location 1, flux was dominated by large 1 mm diameter anchovy fecal pellets. (B) Flux was dominated by small ovoid pellets 200-300 microns in diameter. (C) Location 3 was characterized by very low flux. Flux was dominated by small particles with the occasional large aggregate. (D) Flux was dominated by large aggregates. (E) cumulative normalized volume attenuance vs. equivalent circular diameter curves representative of the 4 locations. Approximately 95% of flux was carried by aggregates >1.5 mm in size at locations 1 and 4. Location 2 had ~50% of flux in >1.5 mm fraction.

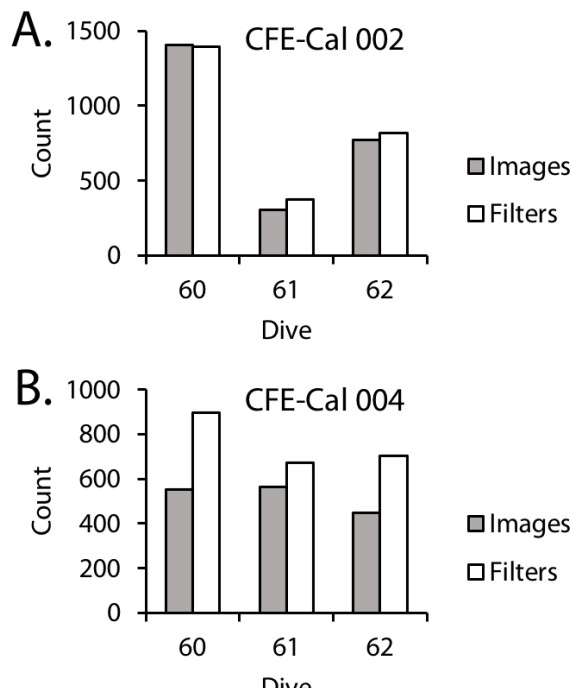

Figure 4: Comparison counts of ovoid pellets in images versus on filters. (A) CFE-CAL002 Deployment 3 (first deployment at location 2) (B) CFE-CAL004 Deployment 4 (second deployment at location 2).

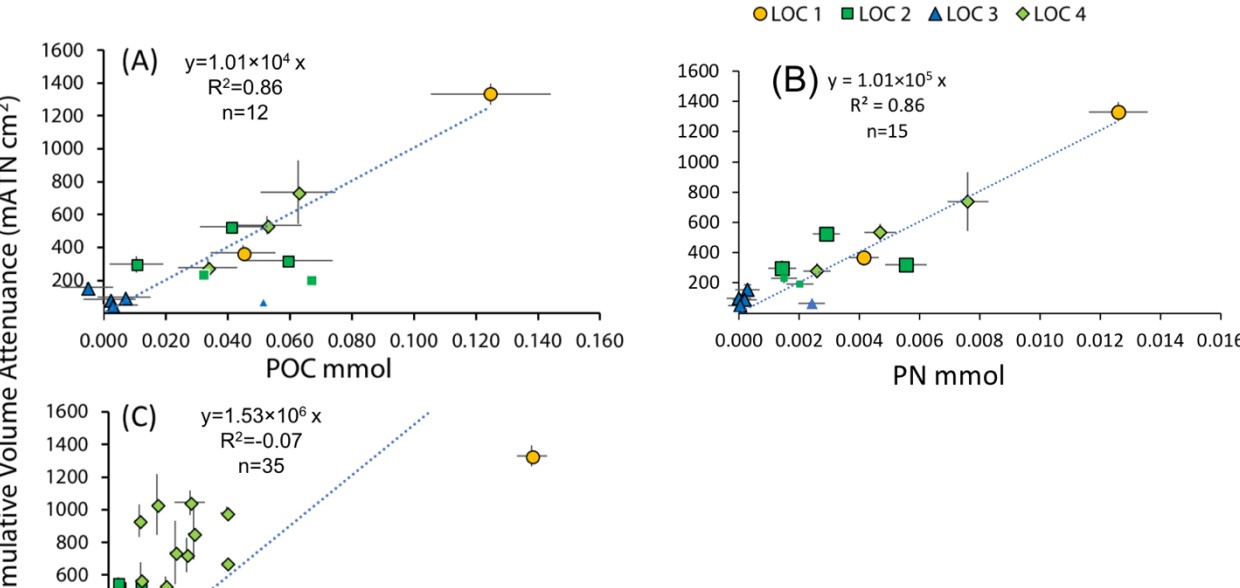

Figure 5: Data and regressions of sample POC (A), PN (B) and PP (C) vs. cumulative volume attenuance. Fits are forced through zero. Smaller symbols in all plots denote samples excluded from the POC regression analysis; these had C/N values >20 and were likely contaminated for carbon and not nitrogen. No data was excluded from PN or PP regressions.

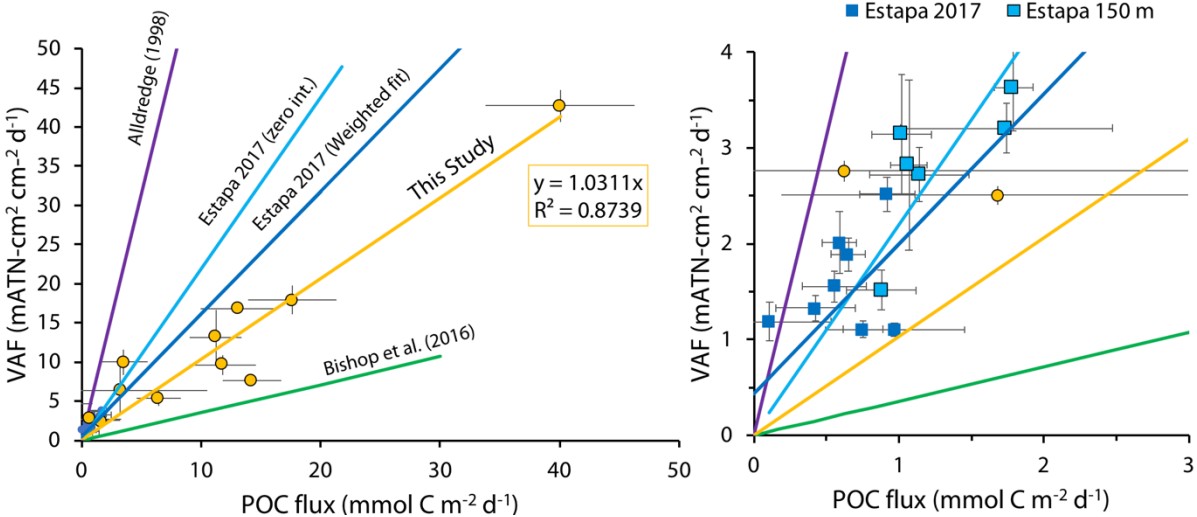

Figure 6: Regressions of Volume Attenuance Flux (mATN-cm$^2$ cm$^{-2}$ d$^{-1}$) to POC (mmol C m$^2$ d$^{-1}$) for this study (orange points and line; y = 1.03x, R$^2$=0.874), Estapa et al. (2017, dark blue, weighted fit: y = 1.56x + 0.434, R$^2$ = 0.632; light blue line, y = 2.191x, R$^2$ = 0.47). Bishop et al. 2016 estimated slope (green line) is y=0.357x (1.0/2.8). Alldredge (1998) estimated slope (purple line) = y=6.25x. As this study's calibration is created using samples collected at 150m, we separate out Estapa's (2017) data points collected in 150m by marking them in light blue for comparison. (A) shows the entire range of VAF and POC flux from this study. (B) expanded graph near the origin (x < 3 mmol C m$^{-2}$ d$^{-1}$) showing the range of Estapa et al. (2017) data.

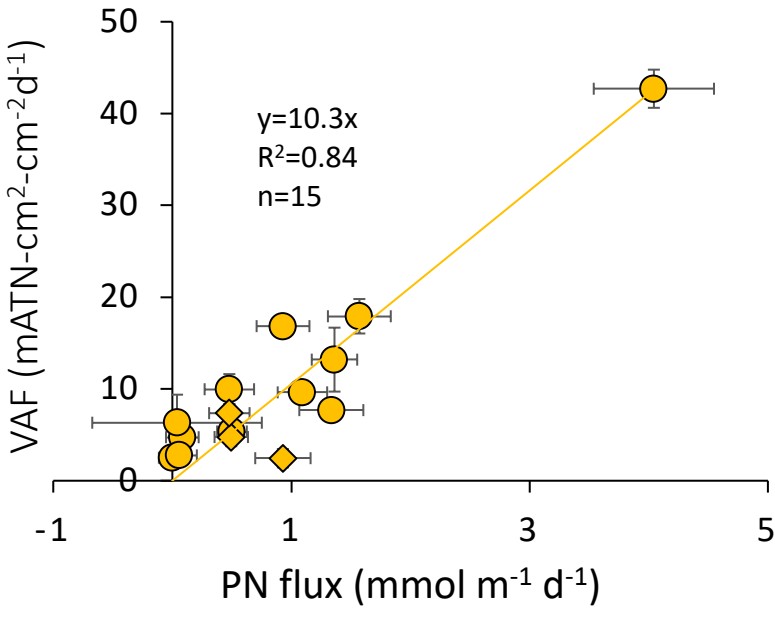

Figure 7: Regressions of Volume Attenuance Flux (mATN-cm$^2$ cm$^{-2}$ d$^{-1}$) to PN (mmol N m$^2$ d$^{-1}$).

**Appendix Figures.**

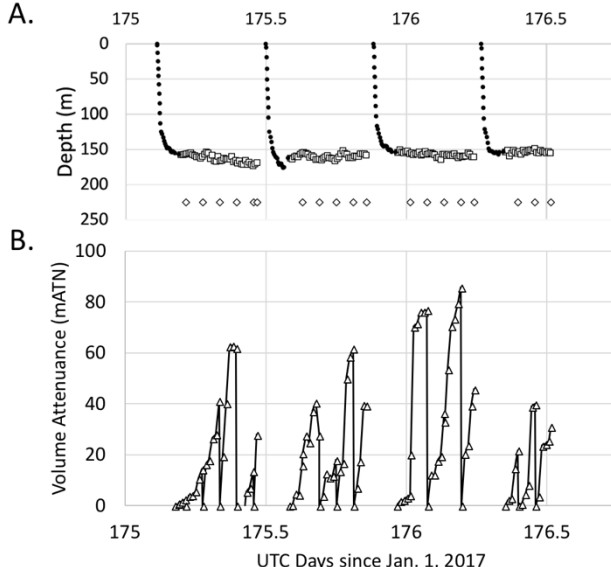

Figure A1: (a) Typical deployment trajectory of a CFE-Cal. This particular deployment is from CFE2, the first deployment at location 4. The x-axis is time in days (Jan 1 2017 at 1200UTC = day 0.5). The filled black circles are depths as the CFE-Cal is diving, open black squares denote depths as the CFE drifts and takes images of settled particles. The open black diamonds represent times when the sample stage was cleaned and particles directed into a sample bottle. (b) Graph B shows the corresponding attenuance for each photo taken. Particles build-up over time and then periodically the glass stage will be rinsed off and particles directed into the sample bottles. Due to a programming error, the sampler and particles are not removed from the stage.

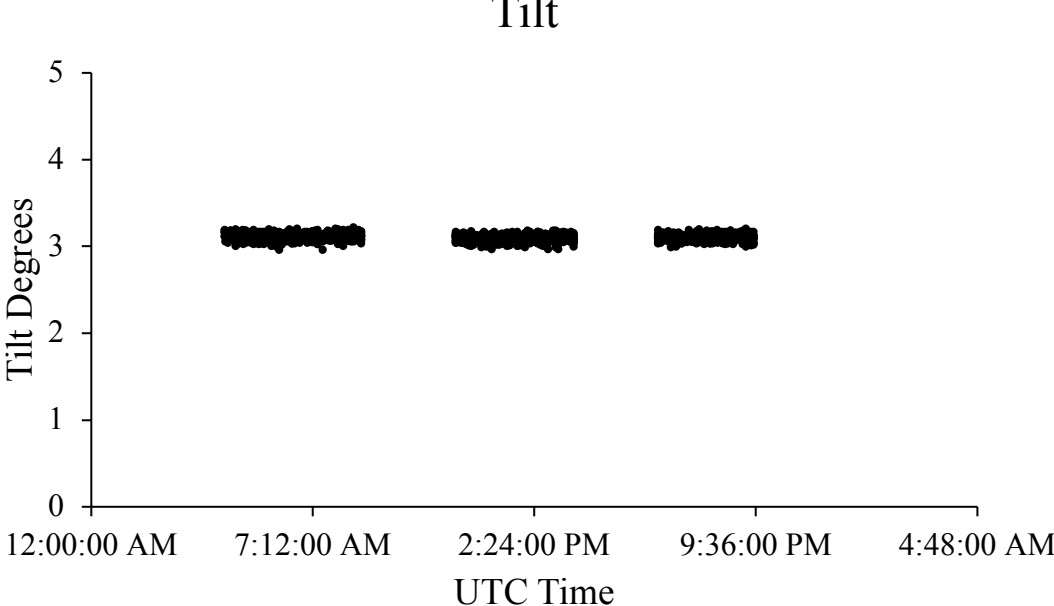

Figure A2: Time series of instrument tilt over the course of a deployment. Data is from CFE-Cal 004's first deployment at location 4. Tilt averages about 3°, consistent with tilt reported for CFEs in Bishop et al., (2016).

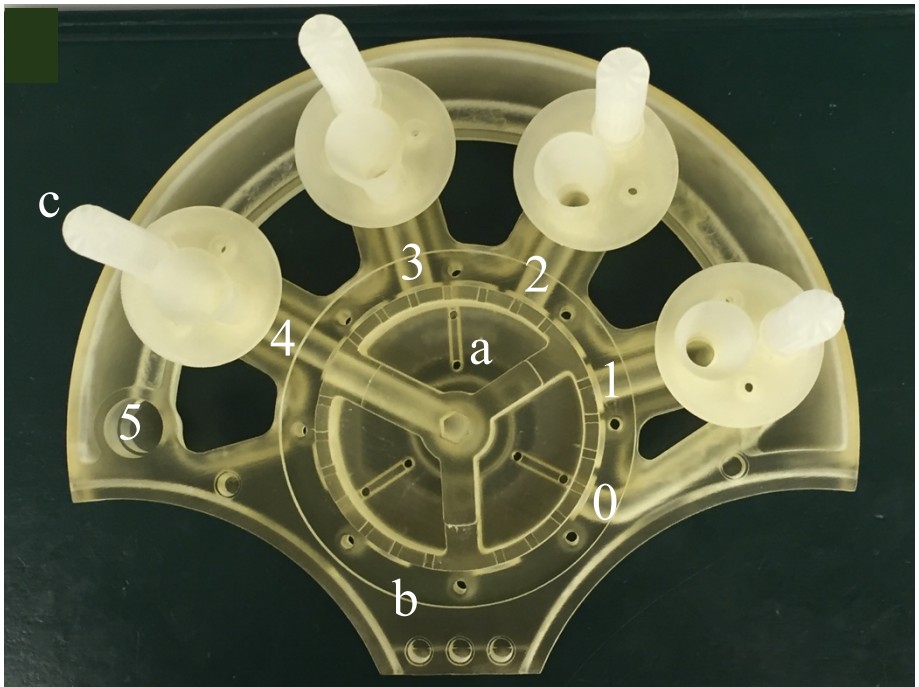

Figure A3: Sampler elements: (a) sample selector rotator; (b) main structural element of the sampler. Flow paths (1-4) direct water and particles into sample bottles or (0) to bypass sample bottles; and (c) particle retention system which bridges inflow channels and common exhaust manifold channel (5). Sample rotator is shown open at position 4. When not sampling, the rotator is sealed to closed positions.

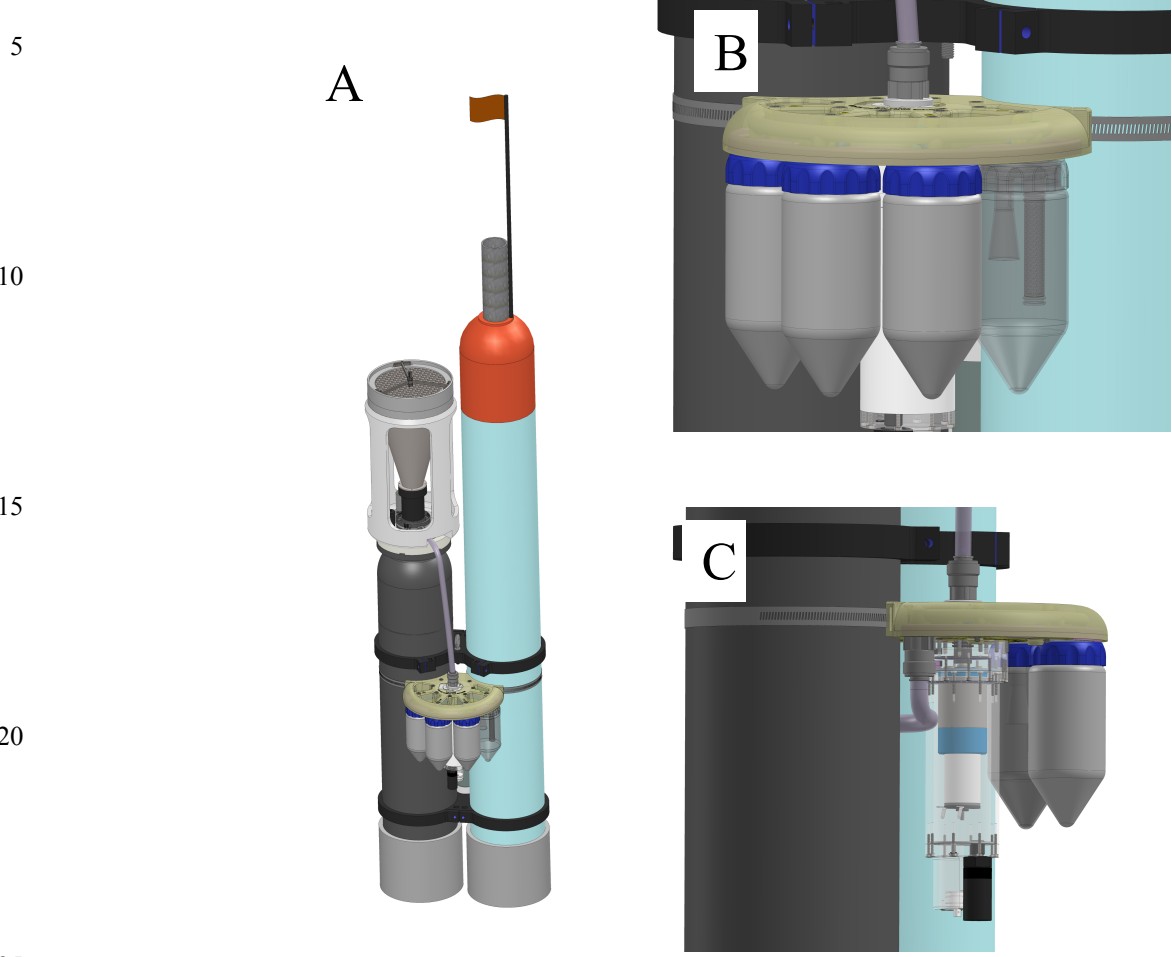

Figure A4: CFE-Cal configuration. (a) CFE-Cal configuration; the CFE-OSR is on the left and the SOLO-2 float is shown on the right. The funnel at the top of the CFE-OSR is covered by hexagonal baffling with 1 cm openings. The funnel is 15.4 cm in diameter. At the base of the funnel is a 2.54 cm diameter glass stage. From here, samples are flushed through the grey tubing pictured above into the sampling unit. (b) Close up of attached sampling system. Far right bottle translucent to show details of bottle inlet and outlet. (c) Side view of sampler with one of the bottles removed to show how motor housing is attached to the sampler. The planetary gear motor is located inside the clear acrylic housing.

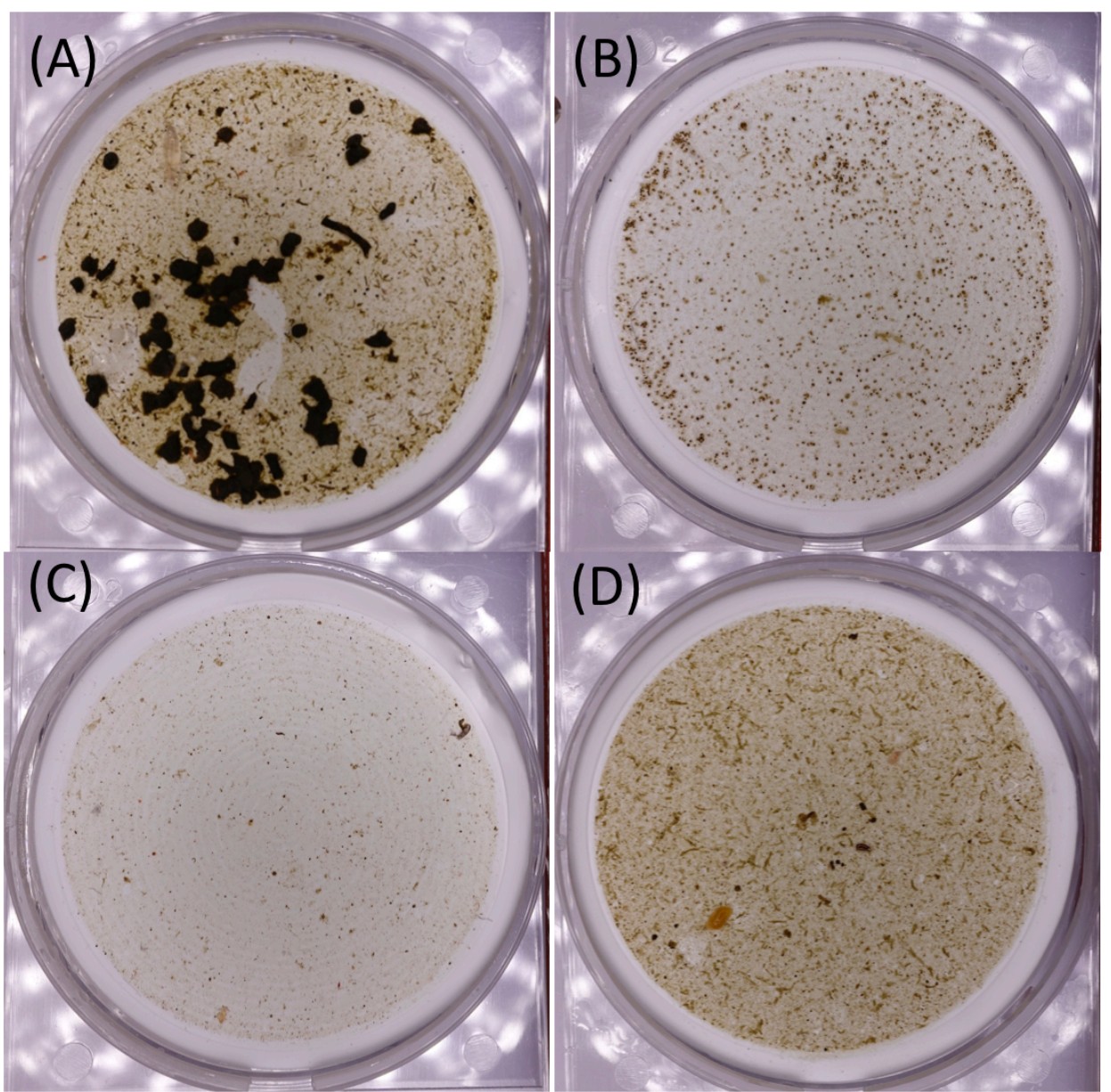

Figure A5: Representative images of sampled particulates from locations 1-4. The process of sampling retains morphology of cohesive aggregates and. Turbulence on transit from imaging stage to bottle does disrupt the integrity of loosely aggregated millimeter sized particles such as represented in Figure 3D. (a) Location 1. CFE 002 dive 42 - Days 160.623 to 160.791 - Depth 119.4 ± 7.8 m. (b) Location 2. CFE 004 dive 71 - Days 167.034 to 167.202 - Depth 157.6 ± 3.4 m. (c) CFE 002 dive 90 – Days 171.190 171.369 - Depth 126.9 ± 4.8 m. (d). Location 4. CFE 002 dive 101 - Days 174.479 to 174.646 - Depth 139.9 ± 3.0 m.