# Peer review of "Carbon Flux Explorer Optical Assessment of C, N and P Fluxes"

_Biogeosciences, 2018_

## Referee Comment (RC1) · Anonymous Referee #1 · 4 Sep 2018

Comments on Bourne et al. paper

This paper conducts quantitative discussion on the relation between beam attenuation and settling particulate organic carbon / nitrogen / phosphate using Carbon Flux Explorer with time-series particle collector (CFE-Cals). In order to study the biological pump for quantifying $CO_2$ transport to the ocean interior, sediment trap experiment has been conducted all over the world ocean. However, moored or surface-tethered or even neutrally buoyant sediment trap has some specific disadvantages such as trapping efficiency and swimmer effect. In addition, it is hard to say that these "cost-performance" is high (need manpower and "ship time"). Nowadays, application of optical sensors such as transmissometer and backscatter meter to the study of marine particulate materials has been becoming more popular. However, although several scientists including one

of co-authors (Prof. Jim Bishop) have been making big efforts to calibrate optical data to actual POC and PIC flux, quantitative conversion of optical data to actual POC data is still on argument because optical observation spatiotemporally synchronized with particle observation has been difficult. Owing to development of CFE-Cals, this study has overcome this problem successfully, succeeding to Estapa et al. (2017). Thus, this paper is valuable for publication. However, I have some question and requests. Especially, discussion on comparison of previous reports is insufficient (explanation of previous papers is ambiguous). I would like to ask authors to make medium revisions as follows.

(Major points) (1) I cannot follow how authors drew Fig.6, especially regression line for previous papers. Please explain how to estimate respective POC: VAF relations of Estapa et al (2017) and Alldredge (1998) (there is no direct description about this relation in the original paper unlike Bishop et al. 2016 (1.0/2.8)) in section 3.4 Comparison to previous studies (or in supplement). (2) According to Figure A4 (Photograph of the surface-tethered BUOY-OSR) in Bishop et al. (2016), it seems that CFE-Cals was installed on the BUOY-OSR. I wonder if this data is not available. Although Bishop et al (2016) concluded that data obtained by BUOY-OSR is underestimated or sampling efficiency is low, if there is data, comparison of optical data and collected settling particle can be possible, POC/ATN relation can be proposed, and comparison of this data and present data can be possible. (3) The configuration figure of CFE-Cals like Figure A1 of Bishop et al (2016) is great helpful for readers to understand CFE-Cals. I strongly recommend authors to add configuration figure of CFE-Cals to this paper. (4) When large amount of settling particle or gigantic settling particle cover over window, settling particle which settle down on covered window cannot be counted and amount of particles or PC must be underestimated with ATN. What do authors think about this?

(Minor points) Page 1 Line 12 (P1L12) Why did not authors measure Ca with ICP-MS? Because Bishop (co-author) reported that "we have no data on the conversion of PIC(POL) to PIC(flux)" in his previous paper (Bishop et al. 2016).

P5L14 Please explain why Fluorinet (3M) was selected as initial liquid.

P5L16 Please explain how to rotate the sample selector rotator (is there motor and gear?)?

P9L20 Insert "(2008)" after "Lamborg et al."

P10L30 Description "(data for regression in Table S1)" should be placed between "this study" and "vs".

P14L18 (reference) C.H.Lamborg => Lamborg, C. H.

Table 1 (1)What does asterisk (*) of some filters mean? Please explain.

(2)I think information of "tilt" is important. How about touching upon information of "tilt" when sampling briefly in table caption or in appropriate place in the text?

Fig. 6 (1) Please explain difference between left figure and right figures. (2) Please explain "Estapa 2017" blue data and "Estapa 150 m " light blue data (150, 300, 500m data set and 150 m data, respectively?) (3) Blue color and light blue color are used not only for different regression lines (forced through zero intercept and allowing for an intercept), but also for different data set (150 m data only and all 150, 300, 500 m data?). This is confusable. Please change color set.

Table S2 (1) No description about Table S2 (2) More detail explanation about respective column in caption

References There are many mistakes and different description (e.g. Deep. Res. => Deep Sea Res, K.O Buesseler <=> Buesseler, K.O.). Please check format.

---

## Referee Comment (RC2) · Anonymous Referee #2 · 30 Oct 2018

**SUMMARY**

In this manuscript the authors address a current critical research field aiming at better estimating the Biological Carbon Pump (BCP) in the ocean by the use of autonomous in situ floats. These devices allow particle flux observations at very high spatio-temporal resolutions essential to capture the rapid ecological changes responsible in a large part for the BCP efficiency variations. In particular, this study targets a calibration between a proxy of particle concentrations in the water column, the volume-attenuance (VA) measured with a Lagrangian float-deployed imaging sediment trap, the Carbon Flux Explorer (CFE), and particle bulk chemical composition in Particulate Organic Carbon (POC), Particulate Nitrogen (PN) and Particulate Phosphorus (PP) measured on the same particles previously imaged and collected with a novel particle sampler

added to the CFE (the whole instrument being named CFE-Cal). The ultimate goal of this calibration is an accurate estimation of element fluxes directly from particle imaging which thus would offer large potential in term of flux data collection which are still today and despite intensive efforts poorly spatially and temporally resolved. After detailing thoroughly the material and methods employed for particle imaging, collection and analyses the authors present results from 15 deployments of the CFE-Cal which lasted 18 to 24 hours near 150m depth in four different locations in the California Current system selected for their contrasting primary productivity features. Results show good correlations between particle content in C and N (but not P) and VA, promising perspectives of using this autonomous in situ imaging to estimate the fluxes of these elements. Each result is discussed (Results and Discussion grouped in the same section) and focus is put on results not meeting authors expectations or not agreeing with the literature. For the results that deviate from expectations, the authors suggest possible explanations from either material malfunctions or the characteristics inherent to the different environment sampled.

GENERAL COMMENTS AND RECOMMENDATIONS

This manuscript is well-written and leaves the reader with the general impression of a solid piece of work. Each section is correctly articulated and information are in general presented where they are expected. Overall, the figures and table shown are clear and deliver well the message intended. The objective tackled here is with no doubt one of the main current and future challenges in BCP research studies (converting particle flux from in situ imaging to biogeochemical fluxes) and I am always pleased to read about work that try to push further our methods to measure these complex and very dynamic ecological processes that drive the BCP with technical innovations. Even if not realising a major advance in the field and presenting results that could be argued, especially in term of potential bias, dataset size and finding significance — this study is worthy being published in BG because it is an attempt to a step forward and will certainly interest the research community working on ocean particle fluxes. However,

even if I acknowledge the work done, its quality and how it is presented I have some concerns about this manuscript that lie mostly on a lack of details about the limitations of the method employed, that also reflect in the results, discussion and the general conclusion made by the authors.

To obtain a good conversion from particle images to POC, PN and PP content, two key parameters have to be carefully considered: (1) the conversion from particle 2D images obtained by the CFE to their 3D volume (detailed in Bishop et al., 2016). Briefly, in Bishop et al. (2016), aggregate (including those of phytodetrital and fecal origin) volume was inferred from cross sectional area converted to equivalent circular diameter and then to volume using an empirical relationship between aggregate thickness and their equivalent circular diameter reported in Bishop et al. (1978); (2) The conversion from particle volume to their chemical content. For that Bishop et al. (2016) used a published value for aggregate dry-weight density (0.087 g cm3; Bishop et al., 1978), and an estimated fraction of organic matter of 60 % in total dry weight. Finally, Bishop et al. (2016) uses an OM:C ratio of 1.88 to convert the estimated OM weight to POC.

The authors highlighted clearly the problem of using these literature-based calibration factors as they are often applicable only in the limited spatio-temporal context of their formulation. But I hardly understand the aim of the present calibration if it is not to finally succeed at reconstructing the flux from images alone and from a large range of environment and ecosystem structures. From the way it is currently presented, the manuscript suggests that the authors are trying to establish a library of relationships between VA and C, N, P contents. If it is the case it should be clearly stated. I would have found very interesting to see in the discussion and conclusion sections some perspectives on how to improve the calibration presented here. In particular, the combined acquisition of particle images and measurements of POC, PN and PP done here offers the great opportunity of estimating the quality of a traditional flux reconstruction (i.e. by inferring its value from the images using published volume to organic contents conversion factors as done before) by comparing it to the real values measured here (as

explored in Estapa et al., 2017). I assume that the final goal is to estimate the POC flux from images alone and much work is yet to be done by the community to understand how to translate small differences in image detection to potentially large differences in chemical contents. A comparison between the calibration method developed here and other methods that try to convert images to elemental fluxes should have been made. The use of Polyacrylamide or Cryogels sediment traps to collect particles and then use image analysis and published values of organic content to convert the images to fluxes is a very close approach to this study. The major advance that the present work could have brought is by extending the use of particle images to push further the estimation of their organic content from the image analysis. It is a bit disappointing to finally re-alise that this study has the great potential of presenting both the images and the "true" values of their content usable to further our understanding of observed discrepancies but that unfortunately this opportunity was not seized by the authors.

Also, the limited number of results obtained due to device malfunction or inherent to the properties of the particle flux collected (i.e. presence of swimmers), or the corrections of POC, PN and PP values obtained from the CFE004 dives due to a discrepancy between images and sampling, should have led the authors to much more caution in their conclusion. Instead, the authors claim "strong calibrations" between the VA and POC-PN contents for a dataset on which many values have been removed or multiplied by an empirically-determined factor; in this case the 1.45 times factor representing the difference of abundance of ovoid pellets in the sampler and from the images.

Based on all these general remarks, I still recommend this article for publication in BG but after significant changes have been made to the Results, Discussion and Conclusion sections and substantial evidences provided where required. In particular, I strongly advise the authors to focus on the general issues mentioned above and summarised as follow: (1) Add to the manuscript a comparison with other techniques of image conversion to biogeochemical fluxes (e.g. gel sediment trap analyses). (2) Use the dataset presented to explore further the known discrepancies between image analysis and inferred organic content. The authors could investigate if a reconstruction of the fluxes measured by the sampler here would be feasible by using the corresponding images and by applying various volume to chemical content relationships to the different particle types identified (e.g. different relationships for fecal pellets, marine snow, etc.). New insights informing on why we struggle at inferring the flux from images would certainly increase significantly the impact that this manuscript will have on the research community. (3) Depending on the modifications made after (1) and (2), moderate if needed the stated significance of the results and discuss it more objectively and into details. Especially, the term "strong correlation" can hardly be used with such confidence knowing that the dataset has been trimmed and partly multiplied by an empirically-determined factor, and that authors seem themselves unsure about potential contaminations of their samples.

Additionally, below are more detailed comments on the manuscript including technical and typographical corrections that will need particular attention before publication. I advise the authors to give a special attention to the four questions/comments below marked with an asterisk (*) as their response should influence the final decision for publication.

DETAILED COMMENTS

Page 1, Line 17: please add to the R2 the size of the sample included in the fit (n) and the p-value. Same line: "...was not sensitive to environment or classes of particles sampled." I assume this statement is used as a proof of applicability of the current calibration to many different ecological contexts. But, it could also suggest that the environment where the deployments were made was not contrasted enough for this calibration.

Page 1, line 21: a space is missing between "Approximately" and "10".

Page 3, line 11: change "our 2.8 conversion factor" to "the 2.8 conversion factor obtained by Bishop et al. (2016)". I understand it is the same team but "our" would mean

a factor inferred in the current study and it is not the case.

Page 4, line 9: the glass stage appears quite small and subjected to overload if a single cm-sized particle (or a few mm-sized particles) happened to enter in the trap. What is the diameter of the opening?

Page 4, line 13: The time of ~25 min seems critical. Is there a threshold at which volume attenuance can be biased by particle overload (particles accumulating over previously deposited particles on the stage)? How did the author choose this time and the time of ~1.8h mentioned below (line 14) as it seems dependent upon the amplitude of the particle flux at the time and depth of the deployment?

Page 6, line 5: do particles larger than the size limit of 3 mm can get stuck inside the openings?

Page 6, lines 7-8: I assume the CFE-Cal has not yet been used for trace metal studies then (this intended use is mentioned above in the manuscript)?

Page 7, line 3: peri slides. Please correct.

Page 7, line 16 and below: why giving results in the Material and Methods section?

* Page 7, lines 19-22: this will need clarification as it seems to be a very serious issue. How could the process blanks be higher than the samples themselves even in case of accidental collection or contamination? Over the 6 replicates of process blanks, how many were contaminated? How did the authors deal with this issue as blanks have to be subtracted from sample values? Are the negative values on Fig. 5A a result of this correction?

Page 7, lines 24-25: "... which we assume is based on sample heterogeneity". Do the authors have evidence to support this assumption?

Page 8, lines 24-25: please mention what would have been the total number of samples in case of no malfunction or swimmers and give a percentage of "fail". My point is

that it is hard to estimate the robustness of the CFE-Cal without a proper estimation of its percentage of fail (how many successful dives/samples over the total number intended?).

Page 8, line 27: this is not really a measure of "collection efficiency" (only assumed) but more a measure of transfer efficiency between the imaging stage and the bottles.

Page 8, line 29: "...close...", please give a precise number.

* Page 8 line 30: again this is a very worrying result that needs more investigation as it suggests a real issue with the collection and/or transfer method employed.

* Page 9, line 1 and lines 3-4: the authors first state that they do not fully understand the issue and then claim to have addressed the problem by solving a software issue. Please bring clarification on this.

Page 9, line 12: what does this time of 2 minutes sample collection time refer to? (how is it different to the $\sim$25 min imaging sequential time?). Is it the duration of particle transfer to the bottles?

Page 9, line 17: please add the sample size (n) and p-values for each regression fits.

Page 9, line 17-25: being "not typical of sinking particles" is certainly not a valid reason to exclude these values from the dataset. Authors are required to provide valid reasons here (e.g. why these C/N ratios would make these particles not wanted in this dataset?).

* Page 9, lines 26-32: again this is a very serious issue. If the sampler building material is potentially responsible for contaminating the samples, how can the authors be confident that not all their POC and PN values are biased by chemicals leaked from this 3D printed part?

Page 10, lines 1-4: this also seems to be pure speculation without any evidence of TEP presence in samples.

Page 10, line 8: if I understand well, the objective of this calibration is to ultimately allow an estimation of biogeochemical fluxes from in situ imaging that could be applied to the largest range of particle types and flux amplitude. It seems very contradictory then to remove particles from the dataset because they are inherently different from the rest of the flux to improve the goodness of the fit. This is very troubling as it suggests that the authors don't fully comprehend their ultimate goal here.

Page 10, line 10-13: this is precisely why it seems so hard to reconstruct a biogeochemical flux from images alone. I strongly suggest that the authors use this example to illustrate the difficulty of meeting the challenge addressed here and impartially discuss their results following the approach of reconstructing the flux from its various particle types having contrasted chemical contents (see general remarks above).

Page 11, line 26: please remove "strong" as it does not seem appropriate. Provide n and p-values.

Page 11, lines 26-27: " that apply over a wide range of environments". This statement could be made with confidence only if the deployments were made in different oceanic regions, seasons and water column layers. It appears too early at this stage to claim this.

Page 11, line 31: "... insensitive to particle classes dominating export". This is not true and is directly contradicted by previous findings shown in this manuscript (see observations made by the authors about the anchovy fecal pellet flux). Please amend as required.

Page 12, lines 5-8: this is confusing and again suggests contradictory intentions of the authors. It is still unclear at this very end of the manuscript if the authors intend to establish a library of VA:element fluxes relationships for each environment and ecological settings sampled (the use of one specific slope would then be reusable to infer the biogeochemical fluxes from images taken in the corresponding region, time of the year and depth), or if they intend to find a general relationship usable in many oceanic

regions, environments and ecosystem structures. In both cases, an extensive work remains to be done and it should be clearly stated.

---

## Author Comment (AC1) · 11 Nov 2018

*Response to Reviewer 1*
Thank you for the prompt review of our paper. We appreciate your insight and the paper will be improved as a result. Below, in black are your review comments and our individualized responses are found in italicized blue. We will address all of the major (I through IV) and minor points (1 through 11) as indicated and hope you find them satisfactory.

**Comments on Bourne et al. paper**

This paper conducts quantitative discussion on the relation between beam attenuation

and settling particulate organic carbon / nitrogen / phosphate using Carbon Flux Explorer with time-series particle collector (CFE-Cals). In order to study the biological pump for quantifying $CO_2$ transport to the ocean interior, sediment trap experiment has been conducted all over the world ocean. However, moored or surface-tethered or even neutrally buoyant sediment trap has some specific disadvantages such as trapping efficiency and swimmer effect. In addition, it is hard to say that these "cost-performance" is high (need manpower and "ship time"). Nowadays, application of optical sensors such as transmissometer and backscatter meter to the study of marine particulate materials has been becoming more popular. However, although several scientists including one of co-authors (Prof. Jim Bishop) have been making big efforts to calibrate optical data to actual POC and PIC flux, quantitative conversion of optical data to actual POC data is still on argument because optical observation spatiotemporally synchronized with particle observation has been difficult. Owing to development of CFE-Cals, this study has overcome this problem successfully, succeeding to Estapa et al. (2017). Thus, this paper is valuable for publication. However, I have some question and requests. Especially, discussion on comparison of previous reports is insufficient (explanation of previous papers is ambiguous). I would like to ask authors to make medium revisions as follows.

**Major Points**

(I) I cannot follow how authors drew Fig.6, especially regression line or previous papers. Please explain how to estimate respective POC: VAF relations of Estapa et al (2017) and Alldredge (1998) (there is no direct description about this relation in the original paper unlike Bishop et al. 2016 (1.0/2.8)) in section 3.4 Comparison to previous studies (or in supplement).

*Estapa calculates attenuance by taking the natural log of transmittance. She reports it in units of ATN m2 m-2 d-1. Our data are log10 transforms of transmittance as documented in Bishop et al. 2016 and reported in units of mATN cm2 cm-2 d-1. Therefore, Estapa's data has been divided by 2.303 to convert the natural log attenuance to log10*

*attenuance and multiplied by 1000 to scale to mATN units. The dimensional data do not require scaling.*

*Bishop et al. 2016 derive the conversion factor for POC:VAF for Alldredge which is 17 times lower than the Bishop et al. 2016 factor of 2.8. Therefore, since axes in the figure are reversed, 2.8/17=.165 compared to our slope of 1. The slope for the Alldredge relationship is 6.*

*We will clarify section 3.4 as suggested.*

(II) According to Figure A4 (Photograph of the surface-tethered BUOY-OSR) in Bishop et al. (2016), it seems that CFE-Cals was installed on the BUOY-OSR. I wonder if this data is not available. Although Bishop et al (2016) concluded that data obtained by BUOY-OSR is underestimated or sampling efficiency is low, if there is data, comparison of optical data and collected settling particle can be possible, POC/ATN relation can be proposed, and comparison of this data and present data can be possible.

*At the time of the 2013 experiments with the BUOY-OSR, we had hoped that analysis of the samples for P scaled by the Redfield ratio would yield carbon. We used Supor filters and analyzed the samples for phosphorous by ICP-MS with this in mind. However, we have shown in this paper that the regression of VA:PP when forced through zero ($r^2$ less than 0) is worse than a random variation around a horizontal line; and only by selecting against the Anchovy pellet dominated sample do we get an $r^2$ of 0.4. In contrast we have an $r^2$= .87 for VA:POC or VA:PN for all samples. The material on Supor (polysulphone) filters cannot be analyzed for POC and PN. Given the fact that phosphorous to attenuance relationship is highly scattered and we do not have POC or PN data, we do not think the BUOY-OSR data are useful to include.*

*We are prepared to mention this in the text, but don't think it adds to the paper.*

(III) The configuration figure of CFE-Cals like Figure A1 of Bishop et al (2016) is great helpful for readers to understand CFE-Cals. I strongly recommend authors to add configuration figure of CFE-Cals to this paper.

*We agree strongly with this reviewer that platform/sampler documentation should be provided. This detail is usually not found. Figure 2 in the main text shows the sampler in its installed context and the flow path through sampler. In the appendix, we provide an image of the key element of the sampler. Figure A2 and its caption describe sampler and sampler flow logic.*

*We will include additional appendix image(s) to provide more detail of the overall configuration and the CFE-Cal sampler.*

(IV) When large amount of settling particle or gigantic settling particle cover over window, settling particle which settle down on covered window cannot be counted and amount of particles or PC must be underestimated with ATN. What do authors think about this?

*As light is reduced exponentially as it passes through particles, as long as the overlapping particles do not 100% obscure the transmitted light, attenuance affects are additive. In our analysis, the transmitted light even in the presence of multiple overlaid large aggregates, never went to zero (in other words, attenuance was never saturating, Bishop et al., 2016). Therefore, overlapping is not an issue.*

*Bishop et al. (2016) discuss the stepwise subtraction of successive attenuance images to derive particle size distribution in the case of overlapped particles (section 2.3). This procedure was required for their estimation of POC to VA. The image set for which this was done is provided in the 2016 supplemental online materials. In the treatment of imagery, we restrict our size-distribution analysis to cases where*

*overlapping effects are minor - this was how the Bishop et al. (2016) SD's were estimated. This had to be done by eye. We are working to implement such an analysis on the CFE.*

**Minor Points**
(1)Page 1 Line 12 (P1L12) Why did not authors measure Ca with ICPMS? Because Bishop (co-author) reported that "we have no data on the conversion of PIC(POL) to PIC(flux)" in his previous paper (Bishop et al. 2016).
*The VA:PP,PN,POC calibration was the priority for this paper and required by our funding. We did measure Ca with the ICPMS for the samples. As the filters had large amounts of residual sea salt, the separation of the non-salt Ca requires very high accuracy and a separate protocol. We are still working on this and the analysis of the cross-polarized light imagery.*

*We do have an accurate and physically based measure of particle birefringence (cross polarized photon yield), which is discussed by Bishop et al., (2016). When a calibration becomes available, all previous results can be translated into units of PIC.*

*We will clarify the text.*

(2) P5L14 Please explain why Fluorinet (3M) was selected as initial liquid. *Fluorinert was selected as it is clear (necessary as there was an optical encoder in the pressure compensated chamber), low viscosity (for motor immersion) and inert (necessary as there were electronics in the chamber). We will clarify this in the text of the Appendix.*

(3) P5L16 Please explain how to rotate the sample selector rotator (is there motor and gear?)?
*There is a motor with a planetary gear set whose output turns the sampler*

(2842S024C; Faulhaber Group, Micromo, Florida). An optical encoder provides feedback as to the proper location for the desired sampling bottle. Included in text.

(4)P9L20 Insert "(2008)" after "Lamborg et al."
*Yes, we will fix this.*

(5) P10L30 Description "(data for regression in Table S1)" should be placed between "this study" and "vs".
*We will move this as suggested.*

(6) P14L18 (reference) C.H.Lamborg => Lamborg, C. H.
*We will fix this as suggested.*

(7) Table 1 (1)What does asterisk (*) of some filters mean? Please explain.
*The Asterix indicates the sample is a blank. Bottom of table indicates this.*

(8) I think information of "tilt" is important. How about touching upon information of "tilt" when sampling briefly in table caption or in appropriate place in the text?
*We will add a figure into the appendix depicting tilt over time, similar to the figure in Bishop et al. (2016). We note that although tilt has negligible effect on particle collection efficiency, our requirement for the minimization of tilt is to facilitate even particle distributions on the sample stage.*

(9) Fig. 6 (1) Please explain difference between left figure and right figures. (2) Please explain "Estapa 2017" blue data and "Estapa 150 m" light blue data (150, 300, 500m data set and 150 m data, respectively?) (3) Blue color and light blue color are used not only for different regression lines (forced through zero intercept and allowing for an intercept), but also for different data set (150 m data only and all 150, 300, 500 m data?). This is confusable. Please change color set.

*We will label the right graph of figure 6 (A) and the left graph (B), then we will reword the caption for figure 6 as follows:*

*Regressions of ATN-POC (mATN-cm$^2$ cm$^{-2}$ d$^{-1}$) to POC (mmol C m$^2$ d$^{-1}$) for this study (orange line; y = 1.03x , R$^2$=0.874), Estapa et al. (2017, blue, y = 1.56x + 0.434, R$^2$ = 0.632; light blue line, y = 2.191x, R$^2$ = 0.47). Bishop et al. 2016 estimated slope (green line) is 0.357 (1.0/2.8). Alldredge (1998) estimated slope (purple line) = 6.25. As this study's calibration is created using samples collected at 150m, we separate out Estapa?s (2017) data point collected in 150m by marking them in light blue for comparison. (A) shows the entire range of VAF and POC flux from this study. (B) expanded graph near the origin (x < 3 mmol C m$^2$ d$^{-1}$) showing the range of Estapa et al. (2017) data.*

(10) Table S2 (1) No description about Table S2 (2) More detail explanation about respective column in caption
*We will add a description of Table S2. Table S2 contains the volume attenuance flux and the POC data for all the samples shown in figure 6. It also notes the CFE-Cal each sample was collected with, the dive number, the bottle number and the length of time the sample was collected over.*

(11) References There are many mistakes and different description (e.g. Deep. Res., K.O Buesseler <=> Buesseler, K.O.). Please check format.
*We will fix all references*

---

## Author Comment (AC2) · 11 Nov 2018

*Response to Reviewer 2*
Thank you for your review. We appreciate your insight. Below, in black are your review comments and our individualized responses are found in italicized blue.

**Comments on Bourne et al. paper**

SUMMARY In this manuscript the authors address a current critical research field aiming at better estimating the Biological Carbon Pump (BCP) in the ocean by the use of autonomous in situ floats. These devices allow particle flux observations at very high spatio-temporal resolutions essential to capture the rapid ecological changes responsible in a large part for the BCP efficiency variations. In particular, this study targets a calibration between a proxy of particle concentrations in the water column, the volume-attenuance (VA) measured with a Lagrangian float-deployed imaging sediment trap, the Carbon Flux Explorer (CFE), and particle bulk chemical composition in Particulate Organic Carbon (POC), Particulate Nitrogen (PN) and Particulate Phosphorus (PP) measured on the same particles previously imaged and collected with a novel particle sampler added to the CFE (the whole instrument being named CFE-Cal). The ultimate goal of this calibration is an accurate estimation of element fluxes directly from particle imaging which thus would offer large potential in term of flux data collection which are still today and despite intensive efforts poorly spatially and temporally resolved. After detailing thoroughly the material and methods employed for particle imaging, collection and analyses the authors present results from 15 deployments of the CFE-Cal which lasted 18 to 24 hours near 150 m depth in four different locations in the California Current system selected for their contrasting primary productivity features. Results show good correlations between particle content in C and N (but not P) and VA, promising perspectives of using this autonomous in situ imaging to estimate the fluxes of these elements. Each result is discussed (Results and Discussion grouped in the same section) and focus is put on results not meeting authors expectations or not agreeing with the literature. For the results that deviate from expectations, the authors suggest possible explanations from either material malfunctions or the characteristics inherent to the different environment sampled.

GENERAL COMMENTS AND RECOMMENDATIONS This manuscript is well-written and leaves the reader with the general impression of a solid piece of work. Each section is correctly articulated and information are in general presented where they are expected. Overall, the figures and table shown are clear and deliver well the message intended. The objective tackled here is with no doubt one of the main current and future challenges in BCP research studies (converting particle flux from in situ imaging to biogeochemical fluxes) and I am always pleased to read about work that try to push further our methods to measure these complex and very dynamic ecological

processes that drive the BCP with technical innovations. Even if not realising a major advance in the field and presenting results that could be argued, especially in term of potential bias, dataset size and finding significance this study is worthy being published in BG because it is an attempt to a step forward and will certainly interest the research community working on ocean particle fluxes. However, even if I acknowledge the work done, its quality and how it is presented I have some concerns about this manuscript that lie mostly on a lack of details about the limitations of the method employed, that also reflect in the results, discussion and the general conclusion made by the authors.

(I) To obtain a good conversion from particle images to POC, PN and PP content, two key parameters have to be carefully considered: (1) the conversion from particle 2D images obtained by the CFE to their 3D volume (detailed in Bishop et al., 2016). Briefly, in Bishop et al. (2016), aggregate (including those of phytodetrital and fecal origin) volume was inferred from cross sectional area converted to equivalent circular diameter and then to volume using an empirical relationship between aggregate thickness and their equivalent circular diameter reported in Bishop et al. (1978); (2) The conversion from particle volume to their chemical content. For that Bishop et al. (2016) used a published value for aggregate dry-weight density (0.087 g cm-3; Bishop et al., 1978), and an estimated fraction of organic matter of 60 % in total dry weight. Finally, Bishop et al. (2016) uses an OM:C ratio of 1.88 to convert the estimated OM weight to POC.

*Following Bishop et al. (2016), our optical results are calculated as Volume attenuance (mAtn-cm2). Bishop et al. (2016) had to estimate POC in the CFE imagery as no calibration samples for POC and PN were collected. In this study, we were able to collect corresponding POC samples using the two CFE-Cals. We therefore did not need to do any conversions of 2D images to 3D volumes to estimate carbon as was done in Bishop et al. (2016). In this paper, we report that the estimate of VA:POC ratio estimated by Bishop et al. using a 2D to 3D volume conversion formula and information on particle density and carbon contents from Bishop et al. (1978), differed by a factor of 3 from the directly calibrated VA:POC relationship. This is shown in*

*figure 6.*

(II) The authors highlighted clearly the problem of using these literature-based calibration factors as they are often applicable only in the limited spatio-temporal context of their formulation. But I hardly understand the aim of the present calibration if it is not to finally succeed at reconstructing the flux from images alone and from a large range of environment and ecosystem structures. From the way it is currently presented, the manuscript suggests that the authors are trying to establish a library of relationships between VA and C, N, P contents. If it is the case it should be clearly stated.

*Yes, our goal is to reconstruct flux from images. In the future, using the CFE-Cals to collect samples from more regions will make the calibration more robust and widely applicable. We will emphasize these goals more clearly.*

(III) I would have found very interesting to see in the discussion and conclusion sections some perspectives on how to improve the calibration presented here. In particular, the combined acquisition of particle images and measurements of POC, PN and PP done here offers the great opportunity of estimating the quality of a traditional flux reconstruction (i.e. by inferring its value from the images using published volume to organic contents conversion factors as done before) by comparing it to the real values measured here (as explored in Estapa et al., 2017). I assume that the final goal is to estimate the POC flux from images alone and much work is yet to be done by the community to understand how to translate small differences in image detection to potentially large differences in chemical contents.

A comparison between the calibration method developed here and other methods that try to convert images to elemental fluxes should have been made. The use of Polyacrylamide or Cryogels sediment traps to collect particles and then use image analysis and published values of organic content to convert the images to fluxes is a very close approach to this study. The major advance that the present work could have

brought is by extending the use of particle images to push further the estimation of their organic content from the image analysis. It is a bit disappointing to finally realise that this study has the great potential of presenting both the images and the "true" values of their content usable to further our understanding of observed discrepancies but that unfortunately this opportunity was not seized by the authors.

*We appreciate the reviewer's suggestion. Such a study of gel trap samples and CFE imagery needs to be done on the same platform and same time and with careful attention to the photography and illumination of gel trap samples. We'd love to facilitate such a comparison in the future but this activity is beyond the scope of this paper. The best same place, same platform, same time study is that of Estapa et al. (2017) who used 2D size distributions in gel trap samples to estimate carbon. They did not extend the analysis of gel trap imagery to attenuance units. The main requirement is that the illumination source intensity (without particles present in the gel) must be mapped precisely as we do with the CFE. Variations in gel homogeneity and thickness degrade particle detection.*

Also, the limited number of results obtained due to device malfunction or inherent to the properties of the particle flux collected (i.e. presence of swimmers), or the corrections of POC, PN and PP values obtained from the CFE004 dives due to a discrepancy between images and sampling, should have led the authors to much more caution in their conclusion.

Instead, the authors claim "strong calibrations" between the VA and POC-PN contents for a dataset on which many values have been removed or multiplied by an empirically-determined factor; in this case the 1.45 times factor representing the difference of abundance of ovoid pellets in the sampler and from the images.

Based on all these general remarks, I still recommend this article for publication in BG but after significant changes have been made to the Results, Discussion and Conclusion sections and substantial evidences provided where required.

In particular, I strongly advise the authors to focus on the general issues mentioned above and summarised as follows:

(A) Add to the manuscript a comparison with other techniques of image conversion to biogeochemical fluxes (e.g. gel sediment trap analyses).
*This recommendation is addressed in (III) above. This is worth-while to do when such particle collections are made simultaneously, ideally on the same platform. We do not agree that comparison to gel trap sample size distribution analysis should be brought into the present discussion.*

(B) Use the dataset presented to explore further the known discrepancies between image analysis and inferred organic content. The authors could investigate if a reconstruction of the fluxes measured by the sampler here would be feasible by using the corresponding images and by applying various volume to chemical content relationships to the different particle types identified (e.g. different relationships for fecal pellets, marine snow, etc.). New insights informing on why we struggle at inferring the flux from images would certainly increase significantly the impact that this manuscript will have on the research community.
*In this paper we show that the VA:POC and VA:PN were largely independent of particle size distributions and class (Figs. 3 and 5). What we meant was that we were impressed by the fact that Anchovy pellet dominated samples fell on the same line for C and N as the amorphous "snow-like" aggregates. We do not think a discussion applying volume-carbon relationships for discrete particle classes to compare with our direct calibration results should be added because (1) such construction of morphologically distinct particle class size distributions from imagery is highly time consuming and (2) models are limited. For these reasons, our group last published on particle flux derived from known particle class concentration size distributions 30 years ago. For example, Bishop et al. (2016) used equations relating aggregate particle size to volume, particle density, and carbon content from particles sampled in the Cape Basin in the South Atlantic (Bishop et al., 1978) to derive particle fluxes in*

[Figure]

*the California coastal waters. We show in this paper that results of predicted VA:POC from Bishop et al. (2016) were a factor of three different then the directly calibrated relationship. We are not sure what volume metric transformations will add to this paper. We are encouraged by our correlations of sample loading and VA.*

(C) Depending on the modifications made after (1) and (2), moderate if needed the stated significance of the results and discuss it more objectively and into details. Especially, the term "strong correlation" can hardly be used with such confidence knowing that the dataset has been trimmed and partly multiplied by an empirically-determined factor, and that authors seem themselves unsure about potential contaminations of their samples.
*These points are addressed below in the detailed comments section.*

Additionally, below are more detailed comments on the manuscript including technical and typographical corrections that will need particular attention before publication. I advise the authors to give a special attention to the four questions/comments below marked with an asterisk (*) as their response should influence the final decision for publication.
DETAILED COMMENTS

1) Page 1, Line 17: please add to the R2 the size of the sample included in the fit (n) and the p-value.
*Yes, here they are below, and they will be added to the manuscript*
*POC, n=13, p=6.0x10$^{-8}$*
*PN, n=16, p=2.0x10$^{-10}$*

2) Same line: "...was not sensitive to environment or classes of particles sampled." I assume this statement is used as a proof of applicability of the current

calibration to many different ecological contexts. But, it could also suggest that the environment where the deployments were made was not contrasted enough for this calibration.

*Figure 3 and 5 were intended to contrast particle size distributions and classes found at each location. We will add clarifications of details of the variability of the environments and size distribution at each collection site.*

3) Page 1, line 21: a space is missing between "Approximately" and "10".
*Yes, will fix.*

4) Page 3, line 11: change "our 2.8 conversion factor" to "the 2.8 conversion factor obtained by Bishop et al. (2016)". I understand it is the same team but "our" would mean version Discussion paper a factor inferred in the current study and it is not the case.
*We will change this.*

5) Page 4, line 9: the glass stage appears quite small and subjected to over-load if a single cm-sized particle (or a few mm-sized particles) happened to enter in the trap. What is the diameter of the opening?
*The diameter of the high aspect ratio funnel is 15.4 cm, the stage diameter is 2.5 cm. The funnel is covered by 1-cm hexagonal light baffle cells. Bishop et al. (2016) provides this full detail. We address sample overloading in the response to point (6).*

6) Page 4, line 13: The time of 25 min seems critical. Is there a threshold at which volume attenuance can be biased by particle overload (particles accumulating over previously deposited particles on the stage)? How did the author choose this time and the time of 1.8h mentioned below (line 14) as it seems dependent upon the amplitude of the particle flux at the time and depth of the deployment?
*The times were chosen to allow for high temporal resolution of flux and since we*

*were not power limited (CFE can operate for 8 months at hourly frequency) we chose by default cleaning times to be the same as times used by Bishop et al. (2016). Overlapping particles do not bias attenuance flux as their contributions are additive. (see discussion in Bishop et al., 2016 and also our response to Reviewer 1 on this point). Implied is the question of whether or not attenuance is saturated (e.g. transmission is 0). The highest fluxes recorded by the CFE were those from January 2013 discussed in Bishop (2016). Even at this high rate of flux, attenuance was not saturated. We have not seen evidence of saturating attenuance in our images from CCE-LTER.*

7) Page 6, line 5: do particles larger than the size limit of 3 mm can get stuck inside the openings?
*Aggregates and rare gelatinous organisms larger than 3 mm do not get stuck. In one of our 23 deployments, a larval crab became stuck in the stage area and had to be removed after recovery. We will add a discussion of this in the text.*

8) Page 6, lines 7-8: I assume the CFE-Cal has not yet been used for trace metal studies then (this intended use is mentioned above in the manuscript)?
*Yes, we designed the CFE-Cal so that in future it can be used for trace metal studies. More refinement is required.*

9) Page 7, line 3: peri slides. Please correct.
*This will be fixed.*

10) Page 7, line 16 and below: why giving results in the Material and Methods section?
*We will move this to the results section.*

11) * Page 7, lines 19-22: this will need clarification as it seems to be a very

serious issue. How could the process blanks be higher than the samples themselves even in case of accidental collection or contamination? Over the 6 replicates of process blanks, how many were contaminated? How did the authors deal with this issue as blanks have to be subtracted from sample values? Are the negative values on Fig. 5A a result of this correction?

*We will modify this section to provide further clarification. We did subtract the blank value from the sample values, as stated in equation 1. This drove one sample negative, though not negative within error. This one sample that was driven negative was collected from location 3. Fluxes at location 3 were very low - an order of magnitude lower than samples collected in other regions. We note that the N for this sample was not negative.*

12) Page 7, lines 24-25: "... which we assume is based on sample heterogeneity". Do the authors have evidence to support this assumption?

*The samples were punched from the filter as described in the methods. We retrieve punches evenly distributed across the filter, but inevitably as there are discrete particles on the filter, there is some heterogeneity between the sub-samples. The sample is not homogenized and then sub-sampled. As this same process is done for every sample, we applied the RSD in the error calculation. We will clarify this in the text. Figure A3 depicts particles dispersed across filters.*

13) Page 8, lines 24-25: please mention what would have been the total number of samples in case of no malfunction or swimmers and give a percentage of "fail". My point is that it is hard to estimate the robustness of the CFE-Cal without a proper estimation of its percentage of fail (how many successful dives/samples over the total number intended?).

*Of the 60 dives, there were 8 dives that failed due to a malfunction (either the float not diving, or the sampler not working). These malfunctions occurred almost entirely at the beginning of the cruise. One dive had a gelatinous swimmer and one had the*

*larval crab. As all instrument malfunctions were resolved, future deployments will be far more robust. We will add a discussion of this to the text.*

14) Page 8, line 27: this is not really a measure of "collection efficiency" (only assumed) but more a measure of transfer efficiency between the imaging stage and the bottles.
*We will change the heading to transfer efficiency.*

15) Page 8, line 29: "...close...", please give a precise number.
*Counts were within error of being the same. We will describe this result statistically.*

16)* Page 8 line 30: again this is a very worrying result that needs more investigation as it suggests a real issue with the collection and/or transfer method employed.
* Page 9, line 1 and lines 3-4: the authors first state that they do not fully understand the issue and then claim to have addressed the problem by solving a software issue. Please bring clarification on this.
*We do understand the issue better than our very brief description implied. We describe the software problem and how we fixed it below.*
*The optical encoder that provides feedback to the motor which turns the sample selector valve was programmed to time out after a certain period of time if the correct "home" position could not be found. During lab tests, where the CFE-Cal was deployed in a large water tank and monitored by video, the sampler would find its correct position in the allowable time. However, these tests were done in a lab, with room temperature water and not in $10°C$ water under pressure.*
*We discovered midway through during deployments that the sampler on CFE4 consistently failed to find the "home" position during pre-deployment checkout. From the "home" position, programming logic advances the sampler by counting encoder pulses to reach open and closed positions for particular sample bottles. This seems to explain the over transfer of particles to this CFE4-Cal. Once this issue was discovered,*

*we increased the time significantly. This issue was not encountered again once it was addressed. We will provide more details on the issue in the text.*

17) Page 9, line 12: what does this time of 2 minutes sample collection time refer to? (how is it different to the 25 min imaging sequential time?). Is it the duration of particle transfer to the bottles?
*This is the duration of particle transfer time during a cleaning cycle. A cleaning cycle takes 2 minutes to complete. The 25-minute timing is the time interval between successive image sets after a cleaning cycle. We will clarify this.*

18) Page 9, line 17: please add the sample size (n) and p-values for each regression fits. Page 9, line 17-25: being "not typical of sinking particles" is certainly not a valid reason to exclude these values from the dataset. Authors are required to provide valid reasons here (e.g. why these C/N ratios would make these particles not wanted in this dataset?).
*We will add regression statistics for all 16 POC points (the 3 excluded ones are already shown in the figure) and report the slope, n and p in the text. See also response to 19 below.*

19) * Page 9, lines 26-32: again this is a very serious issue. If the sampler building material is potentially responsible for contaminating the samples, how can the authors be confident that not all their POC and PN values are biased by chemicals leaked from this 3D printed part?
*Any DOC or DON leaked from the 3D printed part would not be retained in the collected sample. The reason that we excluded the 3 POC points from the regression (see point 18) is that (a) we found visually obvious particles on the filter that were the support material used in the 3D printing process. (b) The C/N of the other 12 samples were all consistent with natural populations. These are in line with C/N that have been found in the region previously (C/N = 11.1 at 100m, Stukel, 2013). As the 3D printed*

*material contains no nitrogen, C/N values would elevated if they were contaminated. In decades of particle collection, Bishop has not found natural material with C:N ratios as high as 20. We will clarify the text.*

20) Page 10, lines 1-4: this also seems to be pure speculation without any evidence of TEP presence in samples.
*This was a hypothesis which we will remove.*

21) Page 10, line 8: if I understand well, the objective of this calibration is to ultimately allow an estimation of biogeochemical fluxes from in situ imaging that could be applied to the largest range of particle types and flux amplitude. It seems very contradictory then to remove particles from the dataset because they are inherently different from the rest of the flux to improve the goodness of the fit. This is very troubling as it suggests that the authors don?t fully comprehend their ultimate goal here. Page 10, line 10-13: this is precisely why it seems so hard to reconstruct a biogeochemical flux from images alone. I strongly suggest that the authors use this example to illustrate the difficulty of meeting the challenge addressed here and impartially discuss their results following the approach of reconstructing the flux from its various particle types having contrasted chemical contents (see general remarks above).
*In the lines you mention, we discuss how the high P content of anchovy fecal pellets, combined with the fast sinking rates, led to phosphorous loadings far higher than the other samples. Though the ultimate goal is to allow an estimation of biogeochemical fluxes based on image analysis, we concluded that we cannot predict particulate phosphorous based on VAF, precisely because of the highly heterogeneous nature of phosphorous in particles, such as the anchovy pellet. In other words, as P is highly labile, we found it was impossible to estimate PP based on in-situ imaging. Our intent was to show that even when eliminating that particular point, the relationship between PP and VAF was still far less robust than that of POC or PN. We will clarify this point*

*more clearly in this section.*

22) Page 11, line 26: please remove "strong" as it does not seem appropriate. Provide n and p-values.
*We will change strong to well-correlated. n and p values are provided.*

23) Page 11, lines 26-27: " that apply over a wide range of environments". This statement could be made with confidence only if the deployments were made in different oceanic regions, seasons and water column layers. It appears too early at this stage to claim this. Page 11, line 31: "... insensitive to particle classes dominating export". This is not true and is directly contradicted by previous findings shown in this manuscript (see observations made by the authors about the anchovy fecal pellet flux). Please amend as required.
*Yes, but we were impressed by the fact that the POC and PN of anchovy pellet dominated samples fell on the same line as the amorphous "snow-like" aggregates. We will modify this to better describe the specific range of environments and particle classes encountered as in P1 Line 17. We will clarify that in the future we hope to make the calibration more robust by further collecting samples from different regions, seasons and depths.*

24) Page 12, lines 5-8: this is confusing and again suggests contradictory intentions of the authors. It is still unclear at this very end of the manuscript if the authors intend to establish a library of VA:element fluxes relationships for each environment and ecological settings sampled (the use of one specific slope would then be reusable to infer the biogeochemical fluxes from images taken in the corresponding region, time of the year and depth), or if they intend to find a general relationship usable in many oceanic regions, environments and ecosystem structures. In both cases, an extensive work remains to be done and it should be clearly stated.
*Our results are a first step towards expanding the range of particle flux that may be*

*retrieved through optical methods. We will rephrase the last sentence. We will also add a discussion of future work to be done.*

---

## Author Response (AR1)

Thank you to the associate editor and the reviewers for your time with this manuscript review. We have addressed all reviewer responses. Below we respond to each reviewers' points and note where relevant changes in the manuscript have been made. We have also made some minor edits to the text for clarity. The one large change we made was we added a figure showing VAF:PN (figure 7), similar to the original VAF:POC figure (figure 6) . The track marked manuscript showing all updates follows the response to reviewers' comments.

*Response to Reviewer 1*

*Thank you for your comments, we appreciate your insight and we believe the paper has been strengthened and improved as a result. Below, we detail how we have updated our manuscript to address your comments. Actions taken are underlined*

**Comments on Bourne et al. paper**

This paper conducts quantitative discussion on the relation between beam attenuation and settling particulate organic carbon / nitrogen / phosphate using Carbon Flux Explorer with time-series particle collector (CFE-Cals). In order to study the biological pump for quantifying $CO_2$ transport to the ocean interior, sediment trap experiment has been conducted all over the world ocean. However, moored or surface-tethered or even neutrally buoyant sediment trap has some specific disadvantages such as trapping efficiency and swimmer effect. In addition, it is hard to say that these "cost-performance" is high (need manpower and "ship time"). Nowadays, application of optical sensors such as transmissometer and backscatter meter to the study of marine particulate materials has been becoming more popular. However, although several scientists including one of co-authors (Prof. Jim Bishop) have been making big efforts to calibrate optical data to actual POC and PIC flux, quantitative conversion of optical data to actual POC data is still on argument because optical observation spatiotemporally synchronized with particle observation has been difficult. Owing to development of CFE-Cals, this study has overcome this problem successfully, succeeding to Estapa et al. (2017). Thus, this paper is valuable for publication. However, I have some question and requests. Especially, discussion on comparison of previous reports is insufficient (explanation of previous papers is ambiguous). I would like to ask authors to make medium revisions as follows.

**Major Points**

(I) I cannot follow how authors drew Fig.6, especially regression line or previous papers. Please explain how to estimate respective POC: VAF relations of Estapa et al (2017) and Alldredge (1998) (there is no direct description about this relation in the original paper unlike Bishop et al. 2016 (1.0/2.8)) in section 3.4 Comparison to previous studies (or in supplement).

*We have added in greater explanation in section 3.4 about how the Estapa et al. (2017) POC:VAF relationship was drived (page 12, line 5-8) and also how the Bishop (1978) and Alldredge (1998) volume:POC relationships were used to derive the POC:VAF relationships presented in Bishop et al. (2016) Added text: page 12 lines 15-32-page 13 line 2.*

(II) According to Figure A4 (Photograph of the surface-tethered BUOY-OSR) in Bishop et al. (2016), it seems that CFE-Cals was installed on the BUOY-OSR. I wonder if this data is not available. Although Bishop et al (2016) concluded that data obtained by BUOY-OSR is underestimated or sampling efficiency is low, if there is data, comparison of optical data and collected settling particle can be possible, POC/ATN relation can be proposed, and comparison of this data and present data can be possible.

*At the time of the 2013 experiments with the BUOY-OSR, we had hoped that analysis of the samples for P scaled by the Redfield ratio would yield carbon. We used Supor filters and analyzed the samples for phosphorous by ICP-MS with this in mind. However, we have shown in this paper that the regression of VA:PP when forced through zero ($r^2<0$) is worse than a random variation around a horizontal line; and only by selecting against the Anchovy pellet dominated sample do we get an $r^2$ of ~0.4. In contrast we have an $r^2= .87$ for VA:POC or VA:PN for all samples. The material on Supor (polysulphone) filters cannot be analyzed for POC and PN. Given the fact that phosphorous to attenuance relationship is highly scattered and we do not have POC or PN data, we do not think the BUOY-OSR data are useful to include.*

*We added a note regarding P analysis of BUOY-OSR samples on Page 3, lines 6 and 7.*

(III) The configuration figure of CFE-Cals like Figure A1 of Bishop et al (2016) is great helpful for readers to understand CFE-Cals. I strongly recommend authors to add configuration figure of CFE-Cals to this paper.

*We agree strongly with this reviewer that platform/sampler documentation should be provided. Figure 2 in the main text shows the sampler in its installed context and the flow path through sampler. In the appendix, we provide an image of the key element of the sampler. Figure A2 and its caption describe sampler and sampler flow logic. We have also included an instrument rendering from the CAD drawings to provide more detail of the overall configuration and the CFE-Cal sampler (supplemental figure A4).*

(IV) When large amount of settling particle or gigantic settling particle cover over window, settling particle which settle down on covered window cannot be counted and amount of particles or PC must be underestimated with ATN. What do authors think about this?

*This is an important consideration.*

*As light is reduced exponentially as it passes through particles, as long as the overlapping particles do not 100% obscure the transmitted light, attenuation affects are additive. In our analysis, the transmitted light even in the presence of multiple overlaid large aggregates, never went to zero (in other words, attenuation was never saturating, Bishop et al., 2016). Therefore, overlapping was not an issue in this study.*

*We added a discussion of this on page 4, line 30 to page 5 line 2.*

**Minor Points**

(1) Page 1 Line 12 (P1L12) Why did not authors measure Ca with ICPMS? Because Bishop (co-author) reported that "we have no data on the conversion of PIC(POL) to PIC(flux)" in his previous paper (Bishop et al. 2016).

*The VA:PP,PN,POC calibration was the priority for this paper and required by our funding. We did measure Ca with the ICPMS for the samples. As the filters had large amounts of residual sea salt, the separation of the non-salt Ca requires very high accuracy and a separate protocol. We are still working on this and the analysis of the cross-polarized light imagery.*

*We do have an accurate and physically based measure of particle birefringence (cross polarized photon yield), which is discussed by Bishop et al., (2016). When a calibration becomes available, all previous results can be translated into units of PIC.*

*We have made a note of this on page 4 lines 12-14.*

(2) P5L14 Please explain why Fluorinet (3M) was selected as initial liquid.

*Fluorinert was selected as it is clear (necessary as there was an optical encoder in the pressure compensated chamber), low viscosity (for motor immersion) and inert (necessary as there were electronics in the chamber).*

*Added discussion to page 5 line 25-27*

(3) P5L16 Please explain how to rotate the sample selector rotator (is there motor and gear?)?

*There is a motor with a planetary gear set whose output turns the sampler (2842S024C; Faulhaber Group, Micromo, Florida). An optical encoder provides feedback as to the proper location for the desired sampling bottle. Included in text.*

*Added details on page 5, lines 22-27.*

(4) P9L20 Insert "(2008)" after "Lamborg et al."

*We fixed this.*

(5) P10L30 Description "(data for regression in Table S1)" should be placed between "this study" and "vs".

*We moved this as suggested.*

(6) P14L18 (reference) C.H.Lamborg => Lamborg, C. H.

*We fixed this as suggested.*

(7) Table 1 (1) What does asterisk (*) of some filters mean? Please explain.

*The Asterix indicates the sample is a process blank. We note this in the figure caption.*

(8) I think information of "tilt" is important. How about touching upon information of "tilt" when sampling briefly in table caption or in appropriate place in the text?

*We added a figure into the appendix (A2) depicting tilt over time, similar to the figure in Bishop et al. (2016). We note that although tilt has negligible effect on particle collection efficiency, our requirement for the minimization of tilt is to facilitate even particle distributions on the sample stage. Added mention of tilt page 5, lines 5 and 6.*

(9) Fig. 6 (1) Please explain difference between left figure and right figures. (2) Please explain "Estapa 2017" blue data and "Estapa 150 m " light blue data (150, 300, 500m data set and 150 m data, respectively?) (3) Blue color and light blue color are used not only for different regression lines (forced through zero intercept and allowing for an intercept), but also for different data set (150 m data only and all 150, 300, 500 m data?). This is confusable. Please change color set.

*We reworded the figure caption as shown below.*

*Regressions of ATN-POC ($mATN$-$cm^2\,cm^{-2}\,d^{-1}$) to POC ($mmol\ C\ m^2\ d^{-1}$) for this study (orange line; $y = 1.03x$ , $R^2=0.874$), Estapa et al. (2017, blue, $y = 1.56x + 0.434$, $R^2 = 0.632$; light blue line, $y = 2.191x$, $R^2 = 0.47$). Bishop et al. 2016 estimated slope (green line) is 0.357 (1.0/2.8). Alldredge (1998) estimated slope (purple line) = 6.25. As this study's calibration is created using samples collected at 150m, we separate out Estapa's (2017) data point collected in 150m by marking them in light blue for comparison. (A) shows the entire range of VAF and POC flux from this study. (B) expanded graph near the origin ($x < 3\ mmol\ C\ m^{-2}\ d^{-1}$) showing the range of Estapa et al. (2017) data.*

(10) Table S2 (1) No description about Table S2 (2) More detail explanation about respective column in caption

*We added a description of Table S2 in the caption. Table S2 contains the volume attenuance flux and the POC data for all the samples shown in figure 6. It also notes the CFE-Cal each sample was collected with, the dive number, the bottle number and the length of time the sample was collected over.*  *Table S2 has been modified.*

(11) References There are many mistakes and different description (e.g. Deep. Res., K.O Buesseler <=> Buesseler, K.O.). Please check format.

10      *We fixed all references*

*Author response to referee 2.*

*Thank you for your review. We appreciate your insight and we believe the paper has been strengthened and improved as a result. Below, we detail how we have updated our manuscript to address your comments. Specific actions are underlined*

SUMMARY In this manuscript the authors address a current critical research field aiming at better estimating the Biological Carbon Pump (BCP) in the ocean by the use of autonomous in situ floats. These devices allow particle flux observations at very high spatio-temporal resolutions essential to capture the rapid ecological changes responsible in a large part for the BCP efficiency variations. In particular, this study targets a calibration between a proxy of particle concentrations in the water

10 column, the volume-attenuance (VA) measured with a Lagrangian float-deployed imaging sediment trap, the Carbon Flux Explorer (CFE), and particle bulk chemical composition in Particulate Organic Carbon (POC), Particulate Nitrogen (PN) and Particulate Phosphorus (PP) measured on the same particles previously imaged and collected with a novel particle sampler added to the CFE (the whole instrument being named CFE-Cal). The ultimate goal of this calibration is an accurate estimation of element fluxes directly from particle imaging which thus would offer large potential in term of flux data collection which

15 are still today and despite intensive efforts poorly spatially and temporally resolved. After detailing thoroughly the material and methods employed for particle imaging, collection and analyses the authors present results from 15 deployments of the CFE-Cal which lasted 18 to 24 hours near 150 m depth in four different locations in the California Current system selected for their contrasting primary productivity features. Results show good correlations between particle content in C and N (but not P) and VA, promising perspectives of using this autonomous in situ imaging to estimate the fluxes of these elements. Each

20 result is discussed (Results and Discussion grouped in the same section) and focus is put on results not meeting authors expectations or not agreeing with the literature. For the results that deviate from expectations, the authors suggest possible explanations from either material malfunctions or the characteristics inherent to the different environment sampled.

GENERAL COMMENTS AND RECOMMENDATIONS This manuscript is well-written and leaves the reader with the
25 general impression of a solid piece of work. Each section is correctly articulated and information are in general presented where they are expected. Overall, the figures and table shown are clear and deliver well the message intended. The objective tackled here is with no doubt one of the main current and future challenges in BCP research studies (converting particle flux from in situ imaging to biogeochemical fluxes) and I am always pleased to read about work that try to push further our methods to measure these complex and very dynamic ecological processes that drive the BCP with technical innovations. Even if not

30 realising a major advance in the field and presenting results that could be argued, especially in term of potential bias, dataset size and finding significance this study is worthy being published in BG because it is an attempt to a step forward and will certainly interest the research community working on ocean particle fluxes. However, even if I acknowledge the work done, its quality and how it is presented I have some concerns about this manuscript that lie mostly on a lack of details about the limitations of the method employed, that also reflect in the results, discussion and the general conclusion made by the authors.

    (I)       To obtain a good conversion from particle images to POC, PN and PP content, two key parameters have to be carefully considered: (1) the conversion from particle 2D images obtained by the CFE to their 3D volume (detailed in Bishop et al., 2016).

            Briefly, in Bishop et al. (2016), aggregate (including those of phytodetrital and fecal origin) volume was

40             inferred from cross sectional area converted to equivalent circular diameter and then to volume using an empirical relationship between aggregate thickness and their equivalent circular diameter reported in Bishop et al. (1978); (2) The conversion from particle volume to their chemical content. For that Bishop et al. (2016) used a published value for aggregate dry-weight density (0.087 g cm$^{-3}$; Bishop et al., 1978), and an estimated fraction of organic matter of 60 % in total dry weight. Finally, Bishop et al. (2016) uses an OM:C ratio of 1.88

45             to convert the estimated OM weight to POC.

*Bishop et al. (2016) had to estimate POC in the CFE imagery as no calibration samples for POC and PN were collected. In this study, we were able to collect corresponding POC samples using the two CFE-Cals. We therefore did not need to do any conversions of 2D images to 3D volumes to estimate carbon as was done in Bishop et al. (2016).*

50

*In this paper, we report that the estimate of VA:POC ratio estimated by Bishop et al. using a 2D to 3D volume conversion formula and information on particle density and carbon contents from Bishop et al. (1978), differed by a factor of 3 from the calibration presented here. This is shown in figure 6. We've added a more detailed explanation of this on page 12 line 15 to page 6 line 6.*

(II)    The authors highlighted clearly the problem of using these literature-based calibration factors as they are often applicable only in the limited spatio-temporal context of their formulation. But I hardly understand the aim of the present calibration if it is not to finally succeed at reconstructing the flux from images alone and from a large range of environment and ecosystem structures. From the way it is currently presented, the manuscript suggests that the authors are trying to establish a library of relationships between VA and C, N, P contents. If it is the case it should be clearly stated.

*Our goal is to reconstruct flux from images. In the future, using the CFE-Cals to collect samples from more regions will make the calibration more robust and widely applicable. We emphasized these goals more clearly in the conclusion. See page 13: lines 18-32 and Page 14 lines 1 to 10.*

(III)   I would have found very interesting to see in the discussion and conclusion sections some perspectives on how to improve the calibration presented here. In particular, the combined acquisition of particle images and measurements of POC, PN and PP done here offers the great opportunity of estimating the quality of a traditional flux reconstruction (i.e. by inferring its value from the images using published volume to organic contents conversion factors as done before) by comparing it to the real values measured here (as explored in Estapa et al., 2017). I assume that the final goal is to estimate the POC flux from images alone and much work is yet to be done by the community to understand how to translate small differences in image detection to potentially large differences in chemical contents.

A comparison between the calibration method developed here and other methods that try to convert images to elemental fluxes should have been made. The use of Polyacrylamide or Cryogels sediment traps to collect particles and then use image analysis and published values of organic content to convert the images to fluxes is a very close approach to this study. The major advance that the present work could have brought is by extending the use of particle images to push further the estimation of their organic content from the image analysis. It is a bit disappointing to finally realise that this study has the great potential of presenting both the images and the "true" values of their content usable to further our understanding of observed discrepancies but that unfortunately this opportunity was not seized by the authors.

*We respond to the suggestion that fluxes should be estimated based on previously published particle volume to carbon relationships as done previously in gel trap studies in response to point B below.*

*In the future, a study of gel trap samples and CFE imagery needs to be done on the same platform and same time and with careful attention to the photography and illumination of gel trap samples. We'd love to facilitate such a comparison in the future but this activity is beyond the scope of this paper. The best same place, same platform, same time study is that of Estapa et al. (2017) who used 2D size distributions in gel trap samples to estimate carbon. They did not extend the analysis of gel trap imagery to attenuation units. The main requirement is that the illumination source intensity (without particles present in the gel) must be mapped precisely as we do with the CFE. Variations in gel homogeneity and thickness degrade particle detection.*

Also, the limited number of results obtained due to device malfunction or inherent to the properties of the particle flux collected (i.e. presence of swimmers), or the corrections of POC, PN and PP values obtained from the CFE004 dives due to a discrepancy between images and sampling, should have led the authors to much more caution in their conclusion.

Instead, the authors claim "strong calibrations" between the VA and POC-PN contents for a dataset on which many values

have been removed or multiplied by an empirically-determined factor; in this case the 1.45 times factor representing the difference of abundance of ovoid pellets in the sampler and from the images.

*We've changed "strong calibrations" to "well-correlated. In the detailed comments section we respond to point 19 on removing the C/N points and point 16 on the 1.45 empirical factor.*

Based on all these general remarks, I still recommend this article for publication in BG but after significant changes have been made to the Results, Discussion and Conclusion sections and substantial evidences provided where required.

In particular, I strongly advise the authors to focus on the general issues mentioned above and summarised as follows:

(A) Add to the manuscript a comparison with other techniques of image conversion to biogeochemical fluxes (e.g. gel sediment trap                                                                                                                                    analyses).

*This recommendation is addressed in (III) above. This is worth-while to do when such particle collections are made simultaneously, ideally on the same platform. We do not agree that comparison to gel trap sample size distribution analysis should be brought into the present discussion.*

(B) Use the dataset presented to explore further the known discrepancies between image analysis and inferred organic content. The authors could investigate if a reconstruction of the fluxes measured by the sampler here would be feasible by using the corresponding images and by applying various volume to chemical content relationships to the different particle types identified (e.g. different relationships for fecal pellets, marine snow, etc.). New insights informing on why we struggle at inferring the flux from images would certainly increase significantly the impact that this manuscript will have on the research community.

*In regards to reconstructing fluxes from published volume to carbon rations, we do compare our VAF:POC to ratios derived using previously published volume to carbon ratios (Alldredge, 1998 and Bishop et al,. 1978) reported in Bishop et al. (2016). Aggregates are a component of flux at all our locations. The typical volume to chemical content relationship for aggregates is from Alldredge (1998). This has been used in both gel trap imagery (Ebersbach and Trull, 2008; Ebersbach et al., 2011) and aggregates collected then imaged using Marine Snow Catchers (Riley et al., 2012; Baker et al., 2017).*

*We've provided a more detailed description of the Alldredge method and why our calibrated relationship of VAF:POC is not consistent with it on page 12 lines 15 to page 13 line 6.*

*The discrepancy likely stems from the fact that aggregates are collected and photographed using different methodologies which may affect their volumes. The Alldredge aggregates are photographed underwater in the plane parallel to the particle sinking direction, whereas the aggregates in the CFE are photographed perpendicular to their sinking direction after the aggregates have settled onto the glass stage. Though settling is very gentle, a 3D particle landing on a 2D surface will have some degree of compaction. Also, the aggregates from the Alldredge (1998) study were collected in the euphotic zone whereas our samples are from the mesopelagic.*

*As we know there are methodological differences between how fluxes are reconstructed from images in gel trap studies and how images are collected using the CFE, and we further know from Bishop et al. (2016) that the relationship of VA:POC derived using the Alldredge (1998) and Bishop et al. (1978) relationships of VAF to POC differ from the relationship directly calibrated in this study, we do not believe it would be useful to add a flux reconstruction based on particle volumes here. We do believe in the future it would be useful to do an intercalibration as mentioned above.*

(C) Depending on the modifications made after (1) and (2), moderate if needed the stated significance of the results and discuss it more objectively and into details. Especially, the term "strong correlation" can hardly be used with such confidence

knowing that the dataset has been trimmed and partly multiplied by an empirically-determined factor, and that authors seem themselves unsure about potential contaminations of their samples.

> *We've modified our conclusion section to stress that these are initial calibration results, and that more work is necessary to make the calibration more robust. We've also removed the term strong correlation, replacing it with well correlated. These points are addressed in further below in the detailed comments section.*

Additionally, below are more detailed comments on the manuscript including technical and typographical corrections that will need particular attention before publication. I advise the authors to give a special attention to the four questions/comments below marked with an asterisk (*) as their response should influence the final decision for publication.

DETAILED COMMENTS

1) Page 1, Line 17: please add to the $R^2$ the size of the sample included in the fit (n) and the p-value.

> *We have added in the n and p values.*

2) Same line: "...was not sensitive to environment or classes of particles sampled." I assume this statement is used as a proof of applicability of the current calibration to many different ecological contexts. But, it could also suggest that the environment where the deployments were made was not contrasted enough for this calibration.

> *Figure 3 and 5 were intended to contrast particle size distributions and classes found at each location. We will add clarifications of details of the variability of the environments and size distribution at each collection site. We have changed Page 1 line 19 to read "was not sensitive to particle size classes or the contrasting environments encountered."*

3) Page 1, line 21: a space is missing between "Approximately" and "10".

> *This has been fixed.*

4) Page 3, line 11: change "our 2.8 conversion factor" to "the 2.8 conversion factor obtained by Bishop et al. (2016)". I understand it is the same team but "our" would mean version Discussion paper a factor inferred in the current study and it is not the case.

> *This was changed.*

5) Page 4, line 9: the glass stage appears quite small and subjected to overload if a single cm-sized particle (or a few mm-sized particles) happened to enter in the trap. What is the diameter of the opening?

> *We added in the diameter of the funnel opening (page 4 line 9). We address sample overloading in the response to point (6).*

6) Page 4, line 13: The time of ~25 min seems critical. Is there a threshold at which volume attenuance can be biased by particle overload (particles accumulating over previously deposited particles on the stage)? How did the author choose this time and the time of ~1.8h mentioned below (line 14) as it seems dependent upon the amplitude of the particle flux at the time and depth of the deployment?

> *The times were chosen to allow for high temporal resolution of flux and since we were not power limited (CFE can operate for 8 months at hourly frequency) we chose by default cleaning times to be the same as times used by Bishop et al. (2016).*

*We have added this discussion to page 4 line 19.*

*Overlapping particles do not bias attenuance flux as their contributions are additive. (see discussion in Bishop et al., 2016 and also our response to Reviewer 1 on this point). Implied is the question of whether or not attenuance is saturated (e.g. transmission is 0). The highest fluxes recorded by the CFE were those from January 2013 discussed in Bishop (2016). Even at this high rate of flux, attenuance was not saturated. We have not seen evidence of saturating attenuance in our images from CCE-LTER.*

*We have added a discussion of this on page 4 line 30 to page 5 line 2.*

7) Page 6, line 5: do particles larger than the size limit of 3 mm can get stuck inside the openings?

*Aggregates and rare gelatinous organisms larger than 3 mm do not get stuck. In one of our 23 deployments, a larval crab became stuck in the stage area and had to be removed after recovery. We added a discussion of this on page 6 line 15-17.*

8) Page 6, lines 7-8: I assume the CFE-Cal has not yet been used for trace metal studies then (this intended use is mentioned above in the manuscript)?

*Yes, we designed the CFE-Cal so that in future it can be used for trace metal studies. More refinement is required. No change to text.*

9) Page 7, line 3: peri slides. Please correct.

*This was fixed.*

10) Page 7, line 16 and below: why giving results in the Material and Methods section?

*This has been moved to the results section.*

11) * Page 7, lines 19-22: this will need clarification as it seems to be a very serious issue. How could the process blanks be higher than the samples themselves even in case of accidental collection or contamination? Over the 6 replicates of process blanks, how many were contaminated? How did the authors deal with this issue as blanks have to be subtracted from sample values? Are the negative values on Fig. 5A a result of this correction?

*We have clarified the discussion of blanks on page 9 lines 9-18.*

*Process blanks were subtracted from sample values, as shown in equation 1. The negative value was a result of this correction. As there were only 5 QMA process blanks, an average of the five was used to blank correct POC and PN. This drove one POC and one PN sample from Location 3 negative, though not negative within error (errors calculated following equation 2). Fluxes at location 3 were very low - an order of magnitude lower than samples collected in other regions.*

*All process blanks and replication (if process blank analysis was replicated, average of the analyses was taken before calculating overall process blank average of the 5 unique samples) had some level of carbon and nitrogen which is attributed to collection either during deployment or sample processing, none were excluded from the calculation.*

12) Page 7, lines 24-25: "... which we assume is based on sample heterogeneity". Do the authors have evidence to support this assumption?

*The samples were punched from the filter as described in the methods. We retrieve punches evenly distributed across the filter, but inevitably as there are discrete particles on the filter, there is some heterogeneity between the sub-samples. The sample is not homogenized and then sub-sampled. As this same process is done for every sample, we applied the RSD in the error calculation. We've more clearly explained this on page 9 line 20-23. Figure A5 depicts particles dispersed across filters.*

13) Page 8, lines 24-25: please mention what would have been the total number of samples in case of no malfunction or swimmers and give a percentage of "fail". My point is that it is hard to estimate the robustness of the CFE-Cal without a proper estimation of its percentage of fail (how many successful dives/samples over the total number intended?).

*We've added a discussion of this to page 8 line21 to page 9 line 2.*

14) Page 8, line 27: this is not really a measure of "collection efficiency" (only assumed) but more a measure of transfer efficiency between the imaging stage and the bottles.

*We changed the Heading 3.2 (page 9, line 26) to Transfer Efficiency.*

15) Page 8, line 29: "...close...", please give a precise number.

*Updated to give percentage on page 8 line2 29-30*
*CFE-Cal2 collected close to the same number of particles in the sampler as were imaged (on average, there was less than a 9% difference between particles imaged and particles collected on the filter, there was not exclusively more in either the CFE images or on the filters as there was for CFE4, see figure 4).*

16) * Page 8 line 30: again this is a very worrying result that needs more investigation as it suggests a real issue with the collection and/or transfer method employed.
* Page 9, line 1 and lines 3-4: the authors first state that they do not fully understand the issue and then claim to have addressed the problem by solving a software issue. Please bring clarification on this.

*We added clarification of the software issue and how it was addressed on page 9 line 31 to page 10 line 6.*

*We do believe that our empirical factor does correctly address the CFE-Cal4 calibration data. We note on page 9 line 19 to 24, that even removing all the CFE-Cal4 data with the collection issue, the VA:POC and VA:PN slope changes less than 5%:*

*We've added the following discussion to page 11 lines 19-24:*

*As mentioned earlier, we found that CFE4 collected 1.45 times more ovoid pellets on the filters than were imaged due to a sampling issue and that we therefore divided the CFE4 POC and PN samples for location 1, 2 and 3 by this empirically derived factor. This affected 6 samples in the POC and PN regressions (see table S1). We note that if instead of applying this empirical factor, these samples are removed from the regression, the VA: POC and VA:PN slopes both change less than 5%. The slopes (number of samples, $R^2$ and p value in parenthesis) change to $10.6 \times 10^3$ (n=8, $R^2$=0.93, p<0.001) and $10.4 \times 10^4$ (n=9, $R^2$=0.91, p<0.001). Using these data which have been corrected using the empirical factor therefore affects the overall regression very little.*

17) Page 9, line 12: what does this time of 2 minutes sample collection time refer to? (how is it different to the ~25 min imaging sequential time?). Is it the duration of particle transfer to the bottles?

*This is the duration of particle transfer time during a cleaning cycle. A cleaning cycle takes 2 minutes to complete. The 25-minute timing is the time interval between successive image sets after a cleaning cycle. We've clarified this on page 10 line 14.*

18) Page 9, line 17: please add the sample size (n) and p-values for each regression fits. Page 9, line 17-25: being "not typical of sinking particles" is certainly not a valid reason to exclude these values from the dataset. Authors are required to provide valid reasons here (e.g. why these C/N ratios would make these particles not wanted in this dataset?).

*We added n and p values page 10 lines 19-20*
*We added the regression values for if the high C/N particles are included on page 11 line 1. We also add more of a discussion of the reasoning behind the C/N exclusion, as discussed below.*

19) * Page 9, lines 26-32: again this is a very serious issue. If the sampler building material is potentially responsible for contaminating the samples, how can the authors be confident that not all their POC and PN values are biased by chemicals leaked from this 3D printed part?

*We've added more discussion of this from page 10 line 27, to page 11 line 2.*

*PN values would not be biased by chemicals from the 3D printed part as the material contains no nitrogen. All the other samples have C/N values typical of sinking particles.*

20) Page 10, lines 1-4: this also seems to be pure speculation without any evidence of TEP presence in samples.

*This was a hypothesis which we removed.*

21) Page 10, line 8: if I understand well, the objective of this calibration is to ultimately allow an estimation of biogeochemical fluxes from in situ imaging that could be applied to the largest range of particle types and flux amplitude. It seems very contradictory then to remove particles from the dataset because they are inherently different from the rest of the flux to improve the goodness of the fit. This is very troubling as it suggests that the authors don't fully comprehend their ultimate goal here. Page 10, line 10-13: this is precisely why it seems so hard to reconstruct a biogeochemical flux from images alone. I strongly suggest that the authors use this example to illustrate the difficulty of meeting the challenge addressed here and impartially discuss their results following the approach of reconstructing the flux from its various particle types having contrasted chemical contents (see general remarks above).

*In the lines you mention, we discuss how the high P content of anchovy fecal pellets, combined with the fast sinking rates, led to phosphorous loadings far higher than the other samples. Though the ultimate goal is to allow an estimation of biogeochemical fluxes based on image analysis, we concluded that we cannot predict particulate phosphorous flux based on VAF, precisely because of the highly heterogeneous nature of phosphorous in particles, such as the anchovy pellet. In other words, as P is highly labile, we found it was impossible to estimate PP based on in-situ imaging. Our intent was to show that even when eliminating that particular point, the relationship between PP and VAF was still far less robust than that of POC or PN.*

*We've removed figure 5 part d, as well as clarified our conclusions on page 11 lines 8 through 17 to make this more clear.*

22) Page 11, line 26: please remove "strong" as it does not seem appropriate. Provide n and p-values.

*We changed strong to well-correlated. N and p values have been added.*

23) Page 11, lines 26-27: " that apply over a wide range of environments". This statement could be made with confidence only if the deployments were made in different oceanic regions, seasons and water column layers. It appears too early at this stage to claim this. Page 11, line 31: "... insensitive to particle classes dominating export". This is not true and is directly contradicted by previous findings shown in this manuscript (see observations made by the authors about the anchovy fecal pellet flux). Please amend as required.

*We changed the wording in the conclusion to describe the different environments encountered.*

24) Page 12, lines 5-8: this is confusing and again suggests contradictory intentions of the authors. It is still unclear at this very end of the manuscript if the authors intend to establish a library of VA:element fluxes relationships for each environment and ecological settings sampled (the use of one specific slope would then be reusable to infer the biogeochemical fluxes from images taken in the corresponding region, time of the year and depth), or if they intend to find a general relationship usable in many oceanic regions, environments and ecosystem structures. In both cases, an extensive work remains to be done and it should be clearly stated.

*Our results are a first step towards expanding the range of particle flux that may be retrieved through optical methods. We've rephrased the conclusion to make this clearer. We also renamed the section "Conclusions and future development", rather than just conclusions. We also stressed multiple times that these are initial results, and more work is necessary to make the calibration more robust.*

[revised manuscript text omitted]

Font: Not Italic, Font color: Text 1

| | | |
|---|---|---|
| **Page 11: [8] Formatted** | **Hannah Bourne** | **11/23/18 12:22:00 PM** |

Font: Not Italic, Font color: Text 1

| | | |
|---|---|---|
| **Page 11: [9] Formatted** | **Hannah Bourne** | **11/23/18 12:22:00 PM** |

Font: Not Italic, Font color: Text 1

| | | |
|---|---|---|
| **Page 11: [9] Formatted** | **Hannah Bourne** | **11/23/18 12:22:00 PM** |

Font: Not Italic, Font color: Text 1

| | | |
|---|---|---|
| **Page 11: [10] Deleted** | **Hannah Bourne** | **12/21/18 10:40:00 AM** |
| **Page 11: [11] Deleted** | **Jim Bishop** | **12/19/18 3:53:00 PM** |

| | | |
|---|---|---|
| **Page 11: [11] Deleted** | **Jim Bishop** | **12/19/18 3:53:00 PM** |

| | | |
|---|---|---|
| **Page 11: [11] Deleted** | **Jim Bishop** | **12/19/18 3:53:00 PM** |

| | | |
|---|---|---|
| **Page 11: [11] Deleted** | **Jim Bishop** | **12/19/18 3:53:00 PM** |

---

## Author Response (AR2)

Thank you very much for the review of our paper. Below we show the updated manuscript including your edits. How we addressed each edit is written in italicized font.

Reviewer 1 Comments Round 2:

Authors addressed respective comments from reviewers appropriately. I think this manuscript has now been improved very well and has become reader-friendly, especially by adding detail explanation of Fig.6 (comparison of VAF:POC relation) in section 3.4 and adding Fig. A4 (CFE-Cal configuration). I'd like to recommend to accept this manuscript to BG after following minor revision.

1. Page 1 Line 19 (P1L19) Need definition of VAF

*We added the definition of VAF*

2. P3L7 Add parenthesis "(" before "as"

*Removed "("*

3. P3LL27 Add ", Fig. A4" after "Fig. 1a"

*Added ", Fig. A4"*

4. P9L27 Add "less than 3 mm" after "ovoid pellets"
*Please describe size of counted ovoid pellets (< 3 mm OK? What is minimum size?)

*Added size range of ovoid pellets*

5. Fig. A4 How about adding some explanation (caption or label) on figure?
(e.g. Sediment trap funnel, Glass sample stage, Major dimension, Planetary gear motor…)

*Added explanation to figure caption.*

[revised manuscript text omitted]